# Fair Sequential Selection Using Supervised Learning Models

**Mohammad Mahdi Khalili**
CIS Department
University of Delaware
Newark, DE, USA
khalili@udel.edu

**Xueru Zhang**
CSE Department
Ohio State University
Columbus, OH, USA
zhang.12807@osu.edu

**Mahed Abroshan**
Alan Turing Institute
London, UK
mabroshan@turing.ac.uk

## Abstract

We consider a selection problem where sequentially arrived applicants apply for a limited number of positions/jobs. At each time step, a decision maker accepts or rejects the given applicant using a pre-trained supervised learning model until all the vacant positions are filled. In this paper, we discuss whether the fairness notions (e.g., equal opportunity, statistical parity, etc.) that are commonly used in classification problems are suitable for the sequential selection problems. In particular, we show that even with a pre-trained model that satisfies the common fairness notions, the selection outcomes may still be biased against certain demographic groups. This observation implies that the fairness notions used in classification problems are not suitable for a selection problem where the applicants compete for a limited number of positions. We introduce a new fairness notion, "Equal Selection (ES)," suitable for sequential selection problems and propose a post-processing approach to satisfy the ES fairness notion. We also consider a setting where the applicants have privacy concerns, and the decision maker only has access to the noisy version of sensitive attributes. In this setting, we can show that the *perfect* ES fairness can still be attained under certain conditions.

## 1 Introduction

Machine learning (ML) techniques have been increasingly used for automated decision-making in high-stake applications such as criminal justice, loan application, face recognition surveillance, etc. While the hope is to improve societal outcomes with these ML models, they may inflict harm by being biased against certain demographic groups. For example, companies such as IBM, Amazon, and Microsoft had to stop sales of their face recognition surveillance technology to the police in summer 2020 because of the significant racial bias [1, 2]. COMPAS (Correctional Offender Management Profiling for Alternative Sanctions), a decision support tool widely used by courts across the United States to predict the recidivism risk of defendants, is biased against African Americans [3]. In lending, the Apple card application system has shown gender biases by assigning a lower credit limit to females than their male counterparts [4].

To measure and remedy the unfairness issues in ML, various fairness notions have been proposed. They can be roughly classified into two categories [5]: 1) **Individual fairness:** it implies that similar individuals should be treated similarly [6, 7, 8]. 2) **Group fairness:** it requires that certain statistical measures to be equal across different groups [9, 10, 11, 12, 13, 14, 15]. In this work, we mainly focus on the notions of group fairness. We consider a sequential selection problem where a set of

35th Conference on Neural Information Processing Systems (NeurIPS 2021).

applicants compete for limited positions and sequentially enter the decision-making system.[1] At each time step, a decision maker accepts or rejects an applicant until $m$ positions are filled. Each applicant can be either qualified or unqualified and has some features related to its qualification state. While applicants' true qualification states are hidden to the decision maker, their features are observable. We assume the decision maker has access to a pre-trained supervised learning model, which maps each applicant's features to a predicted qualification state (qualified or unqualified) or a qualification score indicating the applicant's likelihood of being qualified. Decisions are then made based on these qualification states/scores. Note that this pre-trained model can possibly be biased or satisfy certain group fairness notions (e.g., equal opportunity, statistical parity, etc.).

To make a fair selection with respect to multiple demographic groups, each applicant's group membership (sensitive attribute) is often required. However, in many scenarios, such information can be applicants' private information, and applicants may be concerned about revealing them to the decision maker. As such, we further consider a scenario where instead of the true sensitive attribute, each applicant only reveals a noisy version of the sensitive attribute to the decision maker. We adopt the notion of *local* differential privacy [16] to measure the applicant's privacy. This notion has been widely used by researchers [17, 18, 19] and has been implemented by Apple, Google, Uber, etc.

In this paper, we say the decision is fair if the probability that each position is filled by a qualified applicant from one demographic group is the same as the probability of filling the position by a qualified applicant from the other demographic group. We call this notion equal selection (ES). We first consider the case where the decision maker has access to the applicants' true sensitive attributes. With no limit on the number of available positions (i.e., no competition), our problem can be cast as classification, and statistical parity and equal opportunity constraints are suitable for finding qualified applicants. However, when the number of acceptances is limited (e.g., job application, college admission, award nomination), we can show that the decisions made based on a pre-trained model satisfying statistical parity or equal opportunity fairness may still result in discrimination against a demographic group. It implies that the fairness notions (i.e., statistical parity and equal opportunity) defined for classification problems, are not suitable for sequential selection problems with the limited number of acceptances. We then propose a post-processing method by solving a linear program, which can find a predictor satisfying the ES fairness notion. Our contributions can be summarized as follows,

1. We introduce *Equal Selection* (ES), a fairness notion suitable for the sequential selection problems which ensures diversity among the selected applicants.To the best of our knowledge, this is the first work that studies the fairness issue in sequential selection problems.
2. We show that decisions made based on a pre-trained model satisfying statistical parity or equal opportunity fairness notion may still lead to an unfair and undesirable selection outcome. To address this issue, we use the ES fairness notion and introduce a post-processing approach which solves a linear program and is applicable to any pre-trained model.
3. We also consider a scenario where the applicants have privacy concerns and only report the differentially private version of sensitive attributes. We show that the perfect ES fairness is still attainable even when applicants' sensitive attributes are differentially private.
4. The experiments on real-world datasets validate the theoretical results.

**Related work.** Learning fair supervised machine learning models has been studied extensively in the literature. In general, there are three main approaches to finding a fair predictor,

1. *Pre-processing:* remove pre-existing biases by modifying the training datasets before the training process [20, 21];
2. *In-processing:* impose certain fairness constraint during the training process, e.g., solve a constrained optimization problem or add a regularizer to the objective function [22, 23];
3. *Post-processing:* mitigate biases by changing the output of an existing algorithm [10, 24].

Among fairness constraints, *statistical parity*, *equalized odds*, and *equal opportunity* have gained an increasing attention in supervised learning. Dwork *et al.* [25] studies the relation between individual fairness and statistical parity. They identify conditions under which individual fairness implies statistical parity. Hardt *et al.* in [10] introduce a post-processing algorithm to find an optimal

---

[1]Rolling admission, job application, and award nomination are examples of sequential selection problems with limited approvals/positions. On the other hand, there is no competition in the credit card application as the number of approvals is unlimited.

binary classifier satisfying equal opportunity. Corbett-Davies *et al.* [26] consider the classification in criminal justice with the goal of maximizing public safety subject to a group fairness constraint (e.g., statistical parity, equalized odds, etc.). They show that the optimal policy is in the form of a threshold policy. Cohen *et al.* [27] design a fair hiring policy for a scenario where the employer can set various tests for each candidate and observe a noisy outcome after each test.

There are also works studying both privacy and fairness issues in classification problems. Cummings *et al.* in [28] examine the compatibility of fairness and privacy. They show that it is *impossible* to train a differentially private classifier that satisfies the *perfect* equal opportunity and is more accurate than a constant classifier. This finding leads to several works that design differentially private and approximately fair models [29, 30, 31]. For instance, [29] introduces an algorithm to train a differentially private logistic regression model that is approximately fair. Jagielski *et al.* [30] propose a differentially private fair learning method for training an approximately fair classifier which protects privacy of sensitive attributes. Mozannar *et al.* [31] adopt local differential privacy as the privacy notion and examine the possibility of training a fair classifier given the noisy sensitive attributes that satisfy local differential privacy. In a similar line of research, [32, 33, 34] focus on developing fair models using noisy but not differentially private sensitive attributes. Note that all of the above works assume that the number of acceptances is unlimited (i.e., no competition), and every applicant can be selected as long as it is predicted as qualified.

Our work is closely connected to the literature on selection problems. Unlike classification problems, the number of acceptances is limited in selection problems, and an applicant may not be selected even if it is predicted as qualified. Kleinberg and Raghavan [35] focus on the implicit bias in selection problems and investigate the importance of the *Rooney Rule* in the selection process. They show that this rule effectively improves both the disadvantaged group's representation and the decision maker's utility. Dwork *et al.* [14] also study the selection problem but consider individual fairness notion. Khalili *et al.* [13] study the compatibility of fairness and privacy in selection problems. They use the exponential mechanism and show that it is possible to attain both differential privacy and *perfect* fairness. Note that the selection in all of these works is one shot, i.e., the applicant pool is static, and all the applicants come as a batch but not sequentially.

Fairness in reinforcement learning and online learning is also studied in the literature. In [36], an algorithm is considered fair if it does not prefer an action over another if the long-run reward of the latter is higher than the former. The goal is to learn an optimal long-run policy satisfying such a requirement. Note that this fairness constraint does not apply to classification or selection problems. Joseph *et al.* [37] consider a multi-armed bandit problem with the following fairness notion: Arm $i$ should be selected with higher probability than arm $j$ in round $t$ only if arm $i$ has higher mean reward than arm $j$ in round $t$. This notion is not applicable to our selection problem with a group fairness notion. Metevier *et al.* [38] study an offline multi-armed bandit problem under group fairness notions. However, their method does not address the fairness issue in selection problems because their model does not consider the competition among applicants.

The remainder of the paper is organized as follows. We present our model and introduce the ES fairness in Section 2. We then propose a fair sequential selection algorithm using pre-trained binary classifiers in Section 3. Sequential selection problems using qualification scores are studied in Section 4. We present our numerical experiments in Section 5.

## 2 Model

We consider a sequential selection problem where individuals indexed by $\mathcal{N} = \{1, 2, 3, \ldots\}$ apply for jobs/tasks in a sequential manner. At time step $i$, individual $i$ applies for the task/job and either gets accepted or rejected. The goal of the decision maker is to select $m$ applicants, and s/he continues the process until $m$ applicants get accepted. Each individual $i$ is characterized by tuple $(X_i, A_i, Y_i)$, where $Y_i \in \{0, 1\}$ is the hidden state representing whether individual $i$ is qualified ($Y_i = 1$) for the position or not ($Y_i = 0$), $X_i \in \mathcal{X}$ is the observable feature vector, and $A_i \in \{0, 1\}$ is the sensitive attribute (e.g., gender) that distinguishes an individual's group identity. In this paper, we present our results for a case where $m = 1$. The result can be generalized to $m > 1$ by repeating the same process for $m$ times. We assume tuples $(X_i, A_i, Y_i), i = 1, 2, \ldots$ are i.i.d. random variables following distribution $f_{X,A,Y}(x, a, y)$. For the notational convenience, we sometimes drop index $i$ and use tuple $(X, A, Y)$, which has the same distribution as $(X_i, A_i, Y_i), i = 1, 2, \ldots$.

**Pre-trained supervised learning model.** We assume the decision maker has access to a pre-trained supervised learning model $r : \mathcal{X} \times \{0,1\} \to \mathcal{R} \subseteq [0,1]$ that maps $(X_i, A_i)$ to $R_i = r(X_i, A_i)$, i.e., the predicted qualification state or the qualification score indicating the likelihood of being qualified. In particular, if $\mathcal{R} = \{0,1\}$, then $r(\cdot, \cdot)$ is a binary classifier and $R_i = 0$ (resp. $R_i = 1$) implies that applicant $i$ is predicted as unqualified (resp. qualified); if $\mathcal{R} \neq \{0,1\}$, then $R_i = r(X_i, A_i)$ indicates the qualification score and a higher $R_i$ implies that the individual is more likely to be qualified.

**Selection Procedure.** At time step $i$, an applicant with feature vector $X_i$ and sensitive attribute $A_i$ arrives, and the decision maker uses the output of supervised learning model $r(X_i, A_i)$ to select or reject the individual. If $r(\cdot, \cdot)$ is a binary classifier (i.e., $\mathcal{R} = \{0,1\}$), then the decision maker selects applicant $i$ if $r(X_i, A_i) = 1$ and rejects otherwise. If $\mathcal{R} \neq \{0,1\}$, then $r(X_i, A_i)$ indicates the likelihood of $i$ being qualified and the decision maker uses threshold $\tau \in [0,1]$ to accept/reject the applicant, i.e., accept applicant $i$ if $r(X_i, A_i) \geq \tau$.

**Fairness Metric.** Based on sensitive attribute $A$, the applicants can be divided into two demographic groups. We shall focus on group fairness. Before introducing our algorithm for fair sequential selection, we first define the fairness notion in our setting.

In general, when there is no competition, the fairness notion used in classification problems (e.g., Statistical Parity (SP) [25] and Equal Opportunity (EO) [10]) can improve fairness. However, in our selection problem, where the number of positions is limited, those fairness notions should be adjusted. In particular, as we will see throughout this paper, the decision maker may reject all the applicants from a demographic group while satisfying SP or EO. This shows that EO and SP do not improve diversity among the selected applicants. This motivates us to propose the following fairness notion for (sequential) selection problems, which improves diversity among the selected applicants.

**Definition 1** (Equalized Selection (ES)). *Let $\mathscr{M} : \mathcal{X} \times \{0,1\} \to \{0,1\}$ be an algorithm used by a decision maker at every time step to reject/select an applicant until one applicant is selected. Let $E_a$ denote the event that an applicant with sensitive attribute $A = a$ ($a \in \{0,1\}$) is selected, and $\tilde{Y} = 1$ (resp. $\tilde{Y} = 0$) the event that a qualified (resp. unqualified) applicant is selected under $\mathscr{M}(\cdot)$. Then the selection algorithm $\mathscr{M}(\cdot)$ satisfies Equal Selection (ES)[2] if*

$$\Pr\{E_0, \tilde{Y} = 1\} = \Pr\{E_1, \tilde{Y} = 1\}. \tag{1}$$

To compare the ES notion with SP and EO and understand why SP and EO may lead to undesirable outcomes in a selection problem, consider an example where 100 qualified applicants are competing for three positions. Among them, 90 are from group 0, while ten are from group 1. Under ES, the probability that each position is filled with an applicant from group 0 is the same as that from group 1. Therefore, we expect that the selected applicants are diverse and coming from both groups. However, neither statistical parity nor equal opportunity cares about diversity, and they are likely to result in all the positions being filled by the majority group.

The *perfect* fairness is satisfied when Equation (1) holds. The ES fairness notion can also be relaxed to an approximate version as follows.

**Definition 2** ($\gamma$-Equal Selection). *$\mathscr{M}(\cdot)$ satisfies $\gamma$-Equal Selection ($\gamma$-ES) if*

$$|\Pr\{E_0, \tilde{Y} = 1\} - \Pr\{E_1, \tilde{Y} = 1\}| \leq \gamma. \tag{2}$$

Note that $\gamma \in [0,1]$ quantifies the fairness level, the smaller $\gamma$ implies the fairer selection outcome.

**Accuracy Metric.** Another goal of the decision maker is to maximize the probability of selecting a qualified applicant. Therefore, we define the accuracy of a selection algorithm as follows.

**Definition 3.** *A selection algorithm is $\theta$-accurate if $\Pr(\tilde{Y} = 1) = \theta$.*

Here, $\theta \in [0,1]$ quantifies the accuracy level, and the larger $\theta$ implies the higher accuracy.

---

[2]Similar to EO fairness in classifications, ES fairness only concerns the equity among the qualified applicants. If the qualification of selected applicants doesn't matter, we can consider $\Pr\{E_0\} = \Pr\{E_1\}$ as the fairness notion. While our paper focuses on the notion in Equation (1), all the analysis and results can be extended for $\Pr\{E_0\} = \Pr\{E_1\}$. See Appendix A.1 for more details.

# 3 Fair Selection Using Binary Classifier

## 3.1 Fair selection without privacy guarantee

In this section, we assume that $r(\cdot,\cdot)$ is a binary classifier and $R_i = r(X_i, A_i) \in \{0,1\}$. At time step $t \in \{1, 2, \dots\}$, if $R_t = 1$, then individual $t$ is selected and the decision maker stops the process. Otherwise, the selection process continues until one applicant is being selected. First, we identify a condition under which the perfect ES fairness is satisfied.

**Theorem 1.** *When the pre-trained model $r(\cdot,\cdot)$ is a binary classifier, the perfect ES fairness (1) is satisfied if and only if the following holds,*

$$\Pr\{R = 1, A = 0, Y = 1\} = \Pr\{R = 1, A = 1, Y = 1\}. \tag{3}$$

**Corollary 1.** *If the pre-trained binary classifier $r(\cdot,\cdot)$ satisfies Equal Opportunity fairness defined as $\Pr\{R = 1 | Y = 1, A = 0\} = \Pr\{R = 1 | Y = 1, A = 1\}$ [10], then the selection procedure is perfect ES fair if and only if $\Pr\{A = 0, Y = 1\} = \Pr\{A = 1, Y = 1\}$.*

Note that the condition in Corollary 1 generally does not hold. It shows that equal opportunity (EO) and equal selection (ES) are not compatible with each other.

**ES-Fair Selection Algorithm.** We now introduce a *post-processing* approach to satisfying ES fairness. Suppose a fair predictor $Z \in \{0, 1\}$ is used to accept ($Z = 1$) or reject ($Z = 0$) an applicant. The predictor $Z$ is derived from sensitive attribute $A$ and the output of pre-trained classifier $R = r(X, A)$ based on the following conditional probabilities,

$$\alpha_{a,\hat{y}} := \Pr\{Z = 1 | A = a, R = \hat{y}\}, \hat{y} \in \{0, 1\}, a \in \{0, 1\}.^3$$

Therefore, the fair predictor $Z$ can be found by finding four variables $\alpha_{a,\hat{y}}, \hat{y} \in \{0,1\}, a \in \{0,1\}$. We re-write accuracy $\Pr\{\tilde{Y} = 1\}$ using variables $\alpha_{a,\hat{y}}$ as follows.

$$\Pr\{\tilde{Y} = 1\} = \sum_{i=1}^{\infty} \Pr\{Z_i = 1, Y_i = 1, \{Z_j = 0\}_{j=1}^{i-1}\} = \sum_{i=1}^{\infty} \Pr\{Z_i = 1, Y_i = 1\} \prod_{j=1}^{i-1} \Pr\{Z_j = 0\}$$

$$= \frac{\Pr\{Z = 1, Y = 1\}}{1 - \Pr\{Z = 0\}} = \Pr\{Y = 1 | Z = 1\} = \frac{\sum_{\hat{y},a} \alpha_{a,\hat{y}} \cdot \Pr\{R = \hat{y}, Y = 1, A = a\}}{\sum_{\hat{y},a} \alpha_{a,\hat{y}} \cdot \Pr\{A = a, R = \hat{y}\}},$$

where $\sum_{\hat{y},a} := \sum_{\hat{y} \in \{0,1\}, a \in \{0,1\}}$. To further simplify the notations, denote $\sum_{\hat{y}} := \sum_{\hat{y} \in \{0,1\}}$, $P_{A,R}(a, \hat{y}) := \Pr\{A = a, R = \hat{y}\}$ and $P_{R,Y,A}(\hat{y}, y, a) := \Pr\{R = \hat{y}, Y = y, A = a\}$. Unlike [10], the problem of finding an optimal ES-fair predictor $Z$, which maximizes the accuracy, is a non-linear and non-convex problem. This optimization problem can be written as follows,[4]

$$\max_{\{\alpha_{a,\hat{y}} \in [0,1]\}} \quad \frac{\sum_{\hat{y},a} \alpha_{a,\hat{y}} \cdot P_{R,Y,A}(\hat{y}, 1, a)}{\sum_{\hat{y},a} \alpha_{a,\hat{y}} \cdot P_{A,R}(a, \hat{y})}$$

$$\text{s.t.} \quad (ES) \quad \sum_{\hat{y}} \alpha_{0,\hat{y}} \cdot P_{R,Y,A}(\hat{y}, 1, 0) = \sum_{\hat{y}} \alpha_{1,\hat{y}} \cdot P_{R,Y,A}(\hat{y}, 1, 1). \tag{4}$$

Even though (4) is a non-convex problem, it can be reduced to a linear program below and solved efficiently using the simplex method.

**Theorem 2.** *Assume that $\left[\min_{\hat{y} \in \{0,1\}, a \in \{0,1\}} P_{A,R}(a, \hat{y})\right]$ is not zero. Let $\hat{\alpha}_{a,\hat{y}}, a \in \{0,1\}, \hat{y} \in \{0,1\}$ be the solution to the following linear problem,*

$$\max_{\{\alpha_{a,\hat{y}} \in [0,1]\}} \quad \sum_{\hat{y},a} \alpha_{a,\hat{y}} \cdot P_{R,Y,A}(\hat{y}, 1, a)$$

$$\text{s.t.} \quad (ES) \quad \sum_{\hat{y}} \alpha_{0,\hat{y}} \cdot P_{R,Y,A}(\hat{y}, 1, 0) = \sum_{\hat{y}} \alpha_{1,\hat{y}} \cdot P_{R,Y,A}(\hat{y}, 1, 1),$$

$$\sum_{\hat{y},a} \alpha_{a,\hat{y}} \cdot P_{A,R}(a, \hat{y}) = \min_{\hat{y} \in \{0,1\}, a \in \{0,1\}} P_{A,R}(a, \hat{y}). \tag{5}$$

*Then, $\hat{\alpha}_{a,\hat{y}}, a \in \{0,1\}, \hat{y} \in \{0,1\}$ is the solution to optimization (4). If linear program (5) does not have a solution, then optimization (4) has no solution.*

---

[3]Because $Z$ is derived from $R$ and $A$, $Z$ and $X$ are conditionally independent given $R$ and $A$. Moreover, $Z$ and $Y$ are also conditionally independet given $R$ and $A$.

[4]In Appendix A.2, we will explain how we can write the ES fairness constraint in terms of $\alpha_{a,\hat{y}}$.

Note that the quality (i.e., accuracy) of predictor $Z$ obtained from optimization problem (5) depends on the quality of predictor $R$. Let $Z^*$ be the optimal predictor among all possible predictors satisfying ES fairness (not among the predictors derived from $(A, R)$). In the next theorem, we identify conditions under which the accuracy of $Z$ is close to accuracy of $Z^*$.

**Theorem 3.** *If* $|\Pr\{Y = 1|Z^* = 1\} - \Pr\{Y = 1|R = 1\}| \leq \epsilon$ *and* $\Pr\{A = a|Y = 1, R = 1\} = \Pr\{A = a|R = 1\}, \forall a \in \{0, 1\}$, *then* $|\Pr\{Y = 1|Z^* = 1\} - \Pr\{Y = 1|Z = 1\}| \leq \epsilon$.

Note that $\Pr\{Y = 1|Z^* = 1\}$ and $\Pr\{Y = 1|Z = 1\}$ are accuracy of $Z^*$ and $Z$, respectively. This theorem implies that under certain conditions, if the accuracy of pre-trained model $R$ is sufficiently close to the accuracy of $Z^*$, then the accuracy of predictor $Z$ is also close to the accuracy of $Z^*$.

## 3.2 Fair selection using differentially private sensitive attributes

In this section, we assume the applicants have privacy concerns, and their true sensitive attributes cannot be used directly in the decision-making process.[5] Such a scenario has been studied before in classification problems [30, 31]. We adopt local differential privacy [16] as the privacy measure. Let $\tilde{A}_i \in \{0, 1\}$ be a perturbed version of the true sensitive attribute $A_i$. We say that $\tilde{A}_i$ is $\epsilon$-differentially private if $\frac{\Pr\{\tilde{A}_i=a|A_i=a\}}{\Pr\{\tilde{A}_i=a|A_i=1-a\}} \leq \exp\{\epsilon\}, \forall a \in \{0, 1\}$, where $\epsilon$ is the privacy parameter and sometimes is referred to as the privacy leakage, the larger $\epsilon$ implies a weaker privacy guarantee.

Diffrentially private $\tilde{A}_i$ can be generated using the randomized response algorithm [39], where $\tilde{A}_i$ is generated based on the following distribution,[6]

$$\Pr\{\tilde{A}_i = a|A_i = a\} = \frac{e^\epsilon}{1 + e^\epsilon}, \quad \Pr\{\tilde{A}_i = 1 - a|A_i = a\} = \frac{1}{1 + e^\epsilon}, \quad i \in \{1, 2, \ldots\}. \quad (6)$$

We assume the decision maker does not know the actual sensitive attribute $A_i$ at time step $i$, but has access to the noisy, differentially private $\tilde{A}_i$ generated using the randomized response algorithm. Hence, the decision maker aims to find a set of conditional probabilities $\Pr\{Z = 1|\tilde{A} = \tilde{a}, r(X, \tilde{A}) = \hat{y}\}, \tilde{a} \in \{0, 1\}, \hat{y} \in \{0, 1\}$ to generate a predictor $Z$ that satisfies the ES fairness constraint.

We show in Lemma 1 that even though the true sensitive attribute $A$ is not known to the decision maker at the time of decision-making, the predictor $Z$ derived from $(r(X, \tilde{A}), \tilde{A})$ and the subsequent selection procedure can still satisfy the *perfect* ES fairness. Denote $P_{A,Y,r(X,\tilde{a})}(a, y, \hat{y}) := \Pr\{A = a, Y = y, r(X, \tilde{a}) = \hat{y}\}$ and $P_{r(X,\tilde{a})|A}(\hat{y}|a) := \Pr\{r(X, \tilde{a}) = \hat{y}|A = a\}$ to simplify notations.

**Assumption 1.** *The true sensitive attributes $A$ are included in the training dataset and are available for training function $r(\cdot, \cdot)$. Therefore, $\forall a, \tilde{a}, \hat{y}, \Pr\{A = a, Y = 1, r(X, \tilde{a}) = \hat{y}\}$ is available before the decision making process starts. However, sensitive attribute $A_i$ is not available at time step $i$.*

**Lemma 1.** *Let $\beta_{\tilde{a},\hat{y}} = \Pr\{Z = 1|\tilde{A} = \tilde{a}, r(X, \tilde{A}) = \hat{y}\}$. Predictor $Z$ derived from $(r(X, \tilde{A}), \tilde{A})$ satisfies the ES fairness notion if and only if the following holds,*

$$\sum_{\hat{y}} \beta_{0,\hat{y}} \cdot e^\epsilon \cdot P_{A,Y,r(X,0)}(0, 1, \hat{y}) + \sum_{\hat{y}} \beta_{1,\hat{y}} \cdot P_{A,Y,r(X,1)}(0, 1, \hat{y})$$

$$= \sum_{\hat{y}} \beta_{0,\hat{y}} \cdot P_{A,Y,r(X,0)}(1, 1, \hat{y}) + \sum_{\hat{y}} \beta_{1,\hat{y}} \cdot e^\epsilon P_{A,Y,r(X,1)}(1, 1, \hat{y}). \quad (7)$$

A trivial solution satisfying (7) is $\beta_{\tilde{a},\hat{y}} = 0, \tilde{a} \in \{0, 1\}, \hat{y} \in \{0, 1\}$, under which the predictor $Z$ is a constant classifier and assigns 0 to every applicant, i.e., it rejects all the applicants. It is thus essential to make sure that constraint (7) has a feasible point other than $\beta_{\tilde{a},\hat{y}} = 0, \tilde{a} \in \{0, 1\}, \hat{y} \in \{0, 1\}$. The following lemma introduces a sufficient condition under which (7) has a non trivial feasible point.

**Lemma 2.** *There exists a feasible point except $\beta_{\tilde{a},\hat{y}} = 0, \tilde{a} \in \{0, 1\}, \hat{y} \in \{0, 1\}$ that satisfies (7) if*

$$\epsilon > \max_{a \in \{0,1\}} -\ln \Pr\{R = 1, A = a, Y = 1\}. \quad (8)$$

Using Lemma 1, a set of conditional probabilities $\beta_{\tilde{a},\hat{y}}$ for generating the optimal ES-fair predictor $Z$ can be found by the following optimization problem.

$$\max_{\{\beta_{\tilde{a},\hat{y}} \in [0,1]\}} \quad \Pr\{\tilde{Y} = 1\} \quad \text{s.t.} \quad \text{Equation (7)} \quad (9)$$

---

[5] Sometimes the decision-maker cannot use the true sensitive attribute by regulation. Even if there is no regulation, the applicants may be reluctant to provide the true sensitive attribute.

[6] Note that $\tilde{A}$ is derived purely from $A$. As a result, $(X, Y)$ and $\tilde{A}$ are conditional independent given $A$.

While optimization problem (9) is not a linear optimization (see the proof of the next theorem in the appendix), the optimal $\beta_{\tilde{a},\hat{y}}$ can be found by solving the following linear program.

**Theorem 4.** *Assume that* $\min_{\tilde{a},\hat{y}}[P_A(\tilde{a}) \cdot e^\epsilon \cdot P_{r(X,\tilde{a})|A}(\hat{y}|\tilde{a}) + P_A(1-\tilde{a}) \cdot P_{r(X,\tilde{a})|A}(\hat{y}|1-\tilde{a})] > 0.$
*Let* $\hat{\beta}_{\tilde{a},\hat{y}}, \tilde{a} \in \{0,1\}, \hat{y} \in \{0,1\}$ *be the solution to the following optimization problem.*

$$
\max_{\{\beta_{\tilde{a},\hat{y}} \in [0,1]\}} \quad \sum_{\tilde{a},\hat{y}} \beta_{\tilde{a},\hat{y}} \left[ e^\epsilon P_{A,Y,r(X,\tilde{a})}(\tilde{a}, 1, \hat{y}) + P_{A,Y,r(X,\tilde{a})}(1-\tilde{a}, y, \hat{y}) \right]
$$

$$
s.t. \quad \sum_{\tilde{a},\hat{y}} \beta_{\tilde{a},\hat{y}} \left[ P_A(\tilde{a}) \cdot e^\epsilon \cdot P_{r(X,\tilde{a})|A}(\hat{y}|\tilde{a}) + P_A(1-\tilde{a}) \cdot P_{r(X,\tilde{a})|A}(\hat{y}|1-\tilde{a}) \right]
$$

$$
= \min_{\tilde{a},\hat{y}} \left[ P_A(\tilde{a}) \cdot e^\epsilon \cdot P_{r(X,\tilde{a})|A}(\hat{y}|\tilde{a}) + P_A(1-\tilde{a}) \cdot P_{r(X,\tilde{a})|A}(\hat{y}|1-\tilde{a}) \right],
$$

$$
Equation~(7), \tag{10}
$$

*where* $P_A(\tilde{a}) := \Pr\{A = \tilde{a}\}$, $P_A(1-\tilde{a}) := \Pr\{A = 1-\tilde{a}\}$, *and* $\sum_{\tilde{a},\hat{y}} := \sum_{\tilde{a} \in \{0,1\}, \hat{y} \in \{0,1\}}$. *Then,*
$\hat{\beta}_{\tilde{a},\hat{y}}, \tilde{a} \in \{0,1\}, \hat{y} \in \{0,1\}$ *is the solution to optimization* (9). *If linear program* (10) *does not have a solution, then optimization* (9) *has no solution neither.*

## 4 Selection Using Qualification Score

### 4.1 Fair selection without privacy guarantee

In this section, we consider the case where $\mathcal{R} = [0,1]$ and the supervised model $r(\cdot, \cdot)$ generates a qualification score, which indicates an applicant's likelihood of being qualified. The decision maker selects/rejects each applicant based on the qualification score. We consider a common method where the decisions are made based on a threshold rule, i.e., selecting an applicant if its qualification score $R = r(X, A)$ is above a threshold $\tau$. In other words, prediction $Z_\tau$ is derived from $(R, A)$ based on the following, $Z_\tau = \begin{cases} 1 & \text{if } R \geq \tau \\ 0 & \text{o.w.} \end{cases}$. To simplify the notations, denote $F_R(\tau) := \Pr\{R \leq \tau\}$, $F_{R|a}(\tau) := \Pr\{R \leq \tau | A = a\}$, $F_{R|a,y}(\tau) := \Pr\{R \leq \tau | A = a, Y = y\}$ and $P_{A,Y}(a,y) := \Pr\{A = a, Y = y\}$. Then we have,

$$
\Pr\{E_a, \tilde{Y} = 1\} = \sum_{i=1}^{\infty} P_{A,Y}(a,1) \cdot (1 - F_{R|a,1}(\tau)) \cdot (F_R(\tau))^{i-1} = \frac{P_{A,Y}(a,1)(1 - F_{R|a,1}(\tau))}{1 - F_R(\tau)}.
$$

Predictor $Z_\tau$ satisfies the ES fairness notion if and only if,

$$
P_{A,Y}(0,1) \cdot (1 - F_{R|0,1}(\tau)) = P_{A,Y}(1,1) \cdot (1 - F_{R|1,1}(\tau)). \tag{11}
$$

Since a threshold $\tau$ that satisfies (11) may not exist, we use group-dependent thresholds for two demographic groups. Let $\tau_a$ be the threshold used to select an applicant with sensitive attribute $A = a$. Then, ES fairness holds if and only if the following is satisfied,

$$
P_{A,Y}(0,1) \cdot (1 - F_{R|0,1}(\tau_0)) = P_{A,Y}(1,1) \cdot (1 - F_{R|1,1}(\tau_1)). \tag{12}
$$

The decision maker aims to find the optimal thresholds for two groups by maximizing its accuracy subject to fairness constraint (12). Under thresholds $\tau_0$ and $\tau_1$, the accuracy is given by,

$$
\Pr\{\tilde{Y} = 1\} = \frac{P_{A,Y}(0,1)(1 - F_{R|0,1}(\tau_0)) + P_{A,Y}(1,1)(1 - F_{R|1,1}(\tau_1))}{1 - \eta_{\tau_0,\tau_1}},
$$

where $\eta_{\tau_0,\tau_1} = F_{R|0}(\tau_0) \cdot \Pr\{A = 0\} + F_{R|1}(\tau_1) \cdot \Pr\{A = 1\}$. To find the optimal $\tau_0$ and $\tau_1$, the decision maker solves the following optimization problem,

$$
\max_{\tau_a \in [0,1]} \quad \frac{P_{A,Y}(0,1)(1 - F_{R|0,1}(\tau_0)) + P_{A,Y}(1,1)(1 - F_{R|1,1}(\tau_1))}{1 - \eta_{\tau_0,\tau_1}}
$$

$$
s.t. \quad P_{A,Y}(0,1)(1 - F_{R|0,1}(\tau_0)) = P_{A,Y}(1,1)(1 - F_{R|1,1}(\tau_1)). \tag{13}
$$

If $\mathcal{R} = [0,1]$ and the probability density function of $R$ conditional on $A = a$ and $Y = 1$ is *strictly* positive over $[0,1]$, optimization problem (13) can be easily turned into a one-variable optimization over closed interval $[0,1]$ and the fairness constraint can be removed (see Appendix A.3

for more details). An optimization problem over a closed-interval can be solved using the Bayesian optimization approach [40]. In Appendix A.3, we further consider a special case when $R|A, Y$ is uniformly distributed and find the closed form of the optimal thresholds.

When score $R$ is discrete (i.e., $\mathcal{R} = \{\rho_1, \ldots, \rho_{n'}\}$), optimization problem (13) can also be solved easily using exhaustive search with the time complexity of $\mathcal{O}((n')^2)$.

Next, we show that maximizing $\Pr\{\tilde{Y} = 1\}$ subject to the equal opportunity fairness notion can lead to an undesirable outcome.

**Theorem 5.** *Let $Z$ be the predictor derived by thresholding $R \in \{\rho_1, \ldots, \rho_{n'}\}$ using thresholds $\tau_0, \tau_1$. Moreover, assume that accuracy (i.e., $\Pr\{Y = 1|Z = 1\}$) is increasing in $\tau_0$ and $\tau_1$, and $\Pr\{R = \rho_{n'}|A = a, Y = 1\} \leq \gamma, \forall a \in \{0, 1\}$. Then, under $\gamma$-EO (i.e., $|\Pr\{Z = 1|A = 0, Y = 1\} - \Pr\{Z = 1|A = 1, Y = 1\}| \leq \gamma$), one of the following pairs of thresholds is fair optimal.*

- $\tau_0 > \rho_{n'}$ *and* $\tau_1 = \rho_{n'}$ *(in this case, $\Pr\{R \geq \tau_0|A = 0, Y = 1\} = 0$).*
- $\tau_1 > \rho_{n'}$ *and* $\tau_0 = \rho_{n'}$ *(in this case, $\Pr\{R \geq \tau_1|A = 1, Y = 1\} = 0$).*

Theorem 5 implies that under certain conditions, it is optimal to reject all the applicants from a single demographic group under EO fairness notion. In other words, the EO fairness cannot improve diversity among the selected applicants. We can state a similar theorem for the SP fairness notion (See Appendix A.4).

## 4.2 Fair selection using qualification score and private sensitive attributes

Similar to Section 3.2, we consider the case where the decision maker uses differentially private $\tilde{A}$ instead of true sensitive attribute $A$ during the decision-making process. Let $\tilde{Z}$ be the predictor derived from $r(X, \tilde{A})$ and $\tilde{A}$ according to the following, $\tilde{Z} = \begin{cases} 1 & \text{if } r(X, \tilde{A}) \geq \tilde{\tau}_{\tilde{A}} \\ 0 & \text{o.w.} \end{cases}$ , where $\tilde{\tau}_{\tilde{A}} = \tilde{\tau}_0$ if $\tilde{A} = 0$, and $\tilde{\tau}_{\tilde{A}} = \tilde{\tau}_1$ otherwise. Lemma 3 introduces a necessary and sufficient condition under which predictor $\tilde{Z}$ satisfies the perfect ES fairness. Let $\overline{F}_{r(X,\tilde{a}),A,Y}(\tilde{\tau}_{\tilde{a}}, a, y) := \Pr\{r(X, \tilde{a}) \geq \tilde{\tau}_{\tilde{a}}, A = a, Y = y\}$ and $\overline{F}_{r(X,\tilde{a}),A}(\tilde{\tau}_{\tilde{a}}, a) := \Pr\{r(X, \tilde{a}) \geq \tilde{\tau}_{\tilde{a}}, A = a\}$ and $\overline{F}_{r(X,\tilde{a}),A,\tilde{A}}(\tilde{\tau}_{\tilde{a}}, a, \tilde{a}) := \Pr\{r(X, \tilde{a}) \geq \tilde{\tau}_{\tilde{a}}, A = a, \tilde{A} = \tilde{a}\}$.

**Lemma 3.** *Predictor $\tilde{Z}$ satisfies the perfect ES fairness if and only if $\tilde{\tau}_0$ and $\tilde{\tau}_1$ satisfy the following,*

$$e^\epsilon \cdot \overline{F}_{r(X,0),A,Y}(\tilde{\tau}_0, 0, 1) + \overline{F}_{r(X,1),A,Y}(\tilde{\tau}_1, 0, 1) = e^\epsilon \cdot \overline{F}_{r(X,1),A,Y}(\tilde{\tau}_1, 1, 1) + \overline{F}_{r(X,0),A,Y}(\tilde{\tau}_0, 1, 1). \quad (14)$$

Accuracy $\Pr\{\tilde{Y} = 1\}$ can be written as a function of $\tilde{\tau}_0$ and $\tilde{\tau}_1$ (see Section A.5 for details),

$$\begin{aligned} \Pr\{\tilde{Y} = 1\} &= \frac{\Pr\{\tilde{Z} = 1, Y = 1\}}{\Pr\{\tilde{Z} = 1\}} = \frac{\sum_{a,\tilde{a}} \overline{F}_{r(X,\tilde{a}),A,Y,\tilde{A}}(\tilde{\tau}_{\tilde{a}}, a, 1, \tilde{a})}{\sum_{a,\tilde{a}} \overline{F}_{r(X,\tilde{a}),A,\tilde{A}}(\tilde{\tau}_{\tilde{a}}, a, \tilde{a})} \\ &= \frac{e^\epsilon \sum_a \overline{F}_{r(X,a),A,Y}(\tilde{\tau}_a, a, 1) + \sum_a \overline{F}_{r(X,a),A,Y}(\tilde{\tau}_a, 1 - a, 1)}{e^\epsilon \sum_a \overline{F}_{r(X,a),A}(\tilde{\tau}_a, a) + \sum_a \overline{F}_{r(X,a),A}(\tilde{\tau}_a, 1 - a)}, \quad (15) \end{aligned}$$

where $\sum_a := \sum_{a \in \{0,1\}}$ and $\sum_{a,\tilde{a}} := \sum_{a,\tilde{a} \in \{0,1\}}$. Following Lemma 3, the optimal thresholds $\tilde{\tau}_0$ and $\tilde{\tau}_1$ can be found by maximizing accuracy $\Pr\{\tilde{Y} = 1\}$ subject to fairness constraint (14). That is,

$$\max_{\tilde{\tau}_0, \tilde{\tau}_1} \quad \Pr\{\tilde{Y} = 1\} \quad \text{s.t.} \quad \text{Equation (14).} \quad (16)$$

Similar to optimization (13), if $\mathcal{R} = \{\rho_1, \rho_2, \ldots, \rho_{n'}\}$, then solution to (16) can be found through the exhaustive search with time complexity $\mathcal{O}((n')^2)$.

# 5 Numerical Example

**FICO credit score dataset [41].**[7] FICO credit scores have been used in the US to determine the creditworthiness of people. The dataset used in this experiment includes credit scores from four

---

[7]The datasets used in this paper do not include any identifiable information. The codes are available *here*.

demographic groups (Asian, White, Hispanic, and Black). Cumulative density function (CDF) $\Pr(R \leq \tau | A = a)$ and non-default rate $\Pr(Y = 1 | R = \tau, A = a)$ of each racial group can be calculated from the empirical data (see [10] for more details). In our experiments, we normalize the credit scores from [350,850] to [0,100] and focus on applicants from White ($A = 0$) and Black ($A = 1$) demographic groups. The sample sizes of the white and black groups in the dataset are 133165 and 18274 respectively. Therefore, we estimate group representations as $\Pr(A = 0) = 1 - \Pr(A = 1) = \frac{133165}{133165+18274} = 0.879$.

Figure 1a illustrates the CDF of FICO scores of qualified (i.e., non-default) applicants from White and Black groups. Since $\Pr\{R \leq \rho | Y = 1, A = 0\}$ is always below $\Pr\{R \leq \rho | Y = 1, A = 1\}$, black qualified (non-default) applicants are likely to be assigned lower scores as compared to the white qualified applicants. Therefore, selecting an applicant based on FICO scores will lead to discrimination against black people. We consider three fairness notions in our sequential selection problem: equal opportunity (EO), statistical parity (SP), and equal selection (ES). We say the selection satisfies $\gamma$-equal opportunity ($\gamma$-EO) if $\left| \Pr\{R \geq \tau_0 | A = 0, Y = 1\} - \Pr\{R \geq \tau_1 | A = 1, Y = 1\} \right| \leq \gamma$. Similarly, we say the decision satisfies $\gamma$-SP if $\left| \Pr\{R \geq \tau_0 | A = 0\} - \Pr\{R \geq \tau_1 | A = 1\} \right| \leq \gamma$.

Table 1 summarizes the selection outcomes under ES, SP, and EO. It shows that the accuracy under EO and SP is almost the same as the accuracy under ES fairness. However, the probability that a qualified person is selected from the Black group (i.e., selection rate of Black) under EO and SP is almost zero. This is because the Black group is the minority group (only 12% of the applicants are black). This issue can be addressed using ES fairness which tries to improve diversity. Notice that the

Table 1: Equal Opportunity (EO) v.s. Statistical Parity (SP) v.s. Equal Selection (ES)

| Fairness metric | $\tau_0$ | $\tau_1$ | $\Pr\{E_0, \tilde{Y} = 1\}$ | $\Pr\{E_1, \tilde{Y} = 1\}$ | Accuracy |
|---|---|---|---|---|---|
| 0.01-EO | 99.5 | 99.5 | 0.990 | 0 | 0.990 |
| 0.001-EO | 99.5 | 99.5 | 0.990 | 0 | 0.990 |
| 0.01-SP | 99.5 | 99.5 | 0.990 | 0 | 0.990 |
| 0.001-SP | 99.5 | 99.5 | 0.990 | 0 | 0.990 |
| 0.01-ES | 98.5 | 84.5 | 0.483 | 0.491 | 0.974 |
| 0.001-ES | 98.0 | 65.0 | 0.483 | 0.483 | 0.966 |

optimal thresholds $\tau_0, \tau_1$ in Table 1 are close to the maximum score 100, especially under EO and SP fairness notions. This is because in optimization (13), we have assumed there is no time constraint for the decision maker to find a qualified applicant (i.e., infinite time horizon), and the selection procedure can take a long time. To make the experiment more practical, we add the following *time constraint* to optimization (13): the probability that no applicant is selected after 100 time steps should be less than $\frac{1}{2}$, i.e.,

$$\Pr\{\text{No one is selected in 100 time steps}\} = \left( \Pr\{R < \tau_A\} \right)^{100}$$
$$= \left( \Pr\{A = 0\} \Pr\{R < \tau_0 | A = 0\} + \Pr\{A = 1\} \Pr\{R < \tau_1 | A = 1\} \right)^{100} \leq 0.5. \quad (17)$$

Table 2 summarizes the results when we add the above condition to (13). By comparing Table 2 with Table 1, we observe that $\Pr\{E_1, \tilde{Y} = 1\}$ slightly increases under EO ans SP fairness after adding time constraint (17). Nonetheless, the probability that a qualified applicant is selected from the black community under EO ans SP fairness is still close to zero. More discussions and numerical results about the time constraint are provided in Appendix A.7.

**Adult income dataset [42].** Adult income dataset contains the information of 48,842 individuals, each individual has 14 features including gender, age, education, race, etc. In this experiment, we consider race (White or Black) as the sensitive attribute. We denote White race by $A = 0$ and Black race by $A = 1$. After removing the points with missing values or with the race other than Black and White, we obtain 41,961 data points, among them 37376 belong to the White group. For each data point, we convert all the categorical features to one-hot vectors. In this experiment, we assume the race is individuals' private information, and we aim to evaluate how performances of different selection algorithms are affected by the privacy guarantee. The goal of the decision maker is to select an individual whose annual income is above $50K$ and ensure the selection is fair.

Table 2: Equal Opportunity (EO) v.s. Statistical Parity (SP) v.s. Equal Selection (ES) after adding time constraint (17)

| Fairness metric | $\tau_0$ | $\tau_1$ | $\Pr\{E_0, \tilde{Y} = 1\}$ | $\Pr\{E_1, \tilde{Y} = 1\}$ | Accuracy |
|---|---|---|---|---|---|
| 0.01-EO | 98.0 | 97.5 | 0.947 | 0.042 | 0.989 |
| 0.001-EO | 98.0 | 97.0 | 0.931 | 0.058 | 0.989 |
| 0.01-SP | 98.0 | 98.0 | 0.976 | 0.013 | 0.989 |
| 0.001-SP | 98.0 | 94.0 | 0.873 | 0.115 | 0.988 |
| 0.01-ES | 98.0 | 65.5 | 0.487 | 0.480 | 0.967 |
| 0.001-ES | 98.0 | 65.0 | 0.483 | 0.483 | 0.966 |

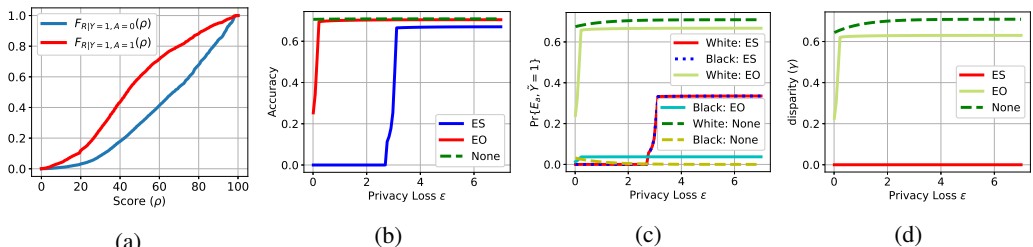

(a)  (b)  (c)  (d)

Figure 1: (a) CDF of FICO scores for non-default applicants from White and Black groups. (b) Accuracy $\Pr\{\tilde{Y} = 1\}$ as a function of $\epsilon$ for the adult dataset. (c) $\Pr\{\tilde{Y} = 1, E_a\}$ of each group. (d) Disparity $\gamma = |\Pr\{E_0, \tilde{Y} = 1\} - \Pr\{E_1, \tilde{Y} = 1\}|$ as a function of $\epsilon$.

We first train a logistic regression classifier (using sklearn package and default parameters) as the pre-trained model. Then for each privacy parameter $\epsilon$, we calculate the probability mass function $P_{\tilde{A}}(\tilde{a})$ using Equation (6). Then, we calculate the joint probability density $P_{A,Y,r(X,\tilde{a})}(a, 1, \hat{y})$ and solve optimization problem (10) to generate a fair predictor for our sequential selection problem. Repeating the process for different privacy loss $\epsilon$, we can find $\Pr\{\tilde{Y} = 1\}, \Pr\{E_0, \tilde{Y} = 1\}, \Pr\{E_1, \tilde{Y} = 1\}$ as a function of privacy loss $\epsilon$. As a baseline, we compare the performance of our algorithm with the following scenarios: 1) *Equal opportunity* (**EO**): replace the ES fairness constraint with the EO constraint in (9) and find the optimal predictor.[8] 2) *No fairness constraint* (**None**): remove the fairness constraint in optimization (9) and find a predictor that maximizes accuracy $\Pr\{\tilde{Y} = 1\}$.

Figure 1b illustrates the accuracy level $\theta = \Pr\{\tilde{Y} = 1\}$ as a function of privacy loss $\epsilon$. Based on Lemma 2, optimization problem (10) has a non-zero solution if $\epsilon$ is larger than a threshold. This is verified in Figure 1b. It shows that if $\epsilon \geq 2.7$, then problem (10) has a non-zero solution. Note that the threshold in Lemma 2 is not tight because $\max_{a \in \{0,1\}} - \ln \Pr\{R = 1, A = a, Y = 1\} = 4.9 > 2.7$. Under ES fairness, accuracy $\Pr\{\tilde{Y} = 1\}$ starts to increase at $\epsilon = 2.7$, and it reaches 0.66 as $\epsilon \to \infty$. Lastly, when $\epsilon \geq 3$, the accuracy under ES fairness is almost the same as that under EO or under no fairness constraint.

Figure 1c illustrates $\Pr\{E_0, \tilde{Y} = 1\}$ and $\Pr\{E_1, \tilde{Y} = 1\}$ as functions of privacy loss. In the case with EO fairness and the case without a fairness constraint, the selection rate of black people always remains close to zero. In contrast, under ES fairness, the available position is filled from Black and White people with the same probability. Figure 1d shows the disparity (i.e., $|\Pr\{E_0, \tilde{Y} = 1\} - \Pr\{E_1, \tilde{Y} = 1\}|$). As expected, disparity remains 0 under ES while it is large in the other cases.

**Limitation and Negative Societal Impact:** 1) We made some assumptions to simplify the problem. For instance, we considered an infinite time horizon and assumed the individuals can be represented by i.i.d. random variables. 2) The proposed fairness notion and the results associated with it are only applicable to our sequential selection problem. This notion may not be suitable for other scenarios.

---

[8]See Appendix A.6 for more details about expressing the EO constraint as a function of $\beta_{\tilde{a}, \hat{y}}$.

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
