# A   Appendix

## A.1   On the ES fairness notion

In this paper, we defined the ES fairness notion as follows,

$$\Pr\{E_0, \tilde{Y} = 1\} = \Pr\{E_1, \tilde{Y} = 1\}$$

Similarly, we could define another fairness notion as follows,

$$\Pr\{E_0\} = \Pr\{E_1\} \tag{18}$$

This definition implies that the probability that a position is filled by an applicant from group $A = 0$ should be the same as the probability that the position is filled by an applicant from group $A = 1$. Consider classifier $R = r(X, A)$. We have,

$$\Pr\{E_a\} = \sum_{i=1}^{\infty} \Pr\{R_i = 1, A_i = a\} \Pr\{R_{i-1} = 0\} \times \cdots \times \Pr\{R_1 = 0\} = \frac{\Pr\{R = 1, A = a\}}{1 - \Pr\{R = 0\}}$$

This implies that (18) is satisfied if and only if

$$\Pr\{R = 1, A = 0\} = \Pr\{R = 1, A = 1\}. \tag{19}$$

Therefore, if equation (18) is our fairness notion, we should design a classifier that satisfies (19). This implies that all the results in this paper can be easily extended to the fairness notion defined in (18).

## A.2   Deriving the ES constraint in terms of $\alpha_{a, \hat{y}}$

Based on Theorem1, $Z$ satisfies the ES if the following holds,

$$\Pr\{A = 0, Z = 1, Y = 1\} = \Pr\{A = 1, Z = 1, Y = 1\} \implies$$

$$\sum_{\hat{y} \in \{0,1\}} \Pr\{A = 0, Z = 1, Y = 1, R = \hat{y}\} = \sum_{\hat{y} \in \{0,1\}} \Pr\{A = 1, Z = 1, Y = 1, R = \hat{y}\} \implies$$

$$\sum_{\hat{y} \in \{0,1\}} \Pr\{Z = 1, Y = 1 | A = 0, R = \hat{y}\} \times \Pr\{A = 0, R = \hat{y}\} =$$

$$\sum_{\hat{y} \in \{0,1\}} \Pr\{Z = 1, Y = 1 | A = 1, R = \hat{y}\} \times \Pr\{A = 1, R = \hat{y}\} \implies$$

$$\sum_{\hat{y} \in \{0,1\}} \alpha_{0, \hat{y}} \Pr\{Y = 1 | A = 0, R = \hat{y}\} \times \Pr\{A = 0, R = \hat{y}\} =$$

$$\sum_{\hat{y} \in \{0,1\}} \alpha_{1, \hat{y}} \Pr\{Y = 1 | A = 1, R = \hat{y}\} \times \Pr\{A = 1, R = \hat{y}\} \implies$$

$$\sum_{\hat{y} \in \{0,1\}} \alpha_{0, \hat{y}} \cdot P_{R,Y,A}(\hat{y}, 1, 0) = \sum_{\hat{y} \in \{0,1\}} \alpha_{1, \hat{y}} \cdot P_{R,Y,A}(\hat{y}, 1, 1).$$

Note that $Z$ and $Y$ are conditionally independent given $A$ and $R$, and we used this fact in the above derivation.

## A.3   On the solution to optimization problem (13)

In this part, first we show that optimization problem (13) can be easily turned into an optimization problem with one variable. If the probability density function of $R$ given $A = a$ and $Y = 1$ is non-zero and continuous over interval $[0, 1]$, then $F_{R|1,1}(.)$ and $F_{R|0,1}(.)$ are both strictly increasing and their inverse functions exits. For the notional convenience, Let $p_{a,y} = P_{A,Y}(a, y)$ and assume $p_{0,1} \leq p_{1,1}$. Then, based on the constraint presented in (13), $\tau_1$ can be calculated as a function of $\tau_0$,

$$\tau_1 = F_{R|1,1}^{-1}\left(1 - \frac{p_{0,1}}{p_{1,1}}(1 - F_{R|0,1}(\tau_0))\right), \tau_0 \in [0, 1]. \tag{20}$$

Therefore, optimization problem (13) can be written as a one-variable optimization problem in interval [0,1] and can be efficiently solved using the Bayesian Optimization algorithm [40].

Now we solve optimization problem (13) for a case where $R|A = a, Y = 1$ follows the uniform distribution.

**Case 1**: consider the following scenario,

- $R|A = 0, Y = 0$ follows $Uniform(0, b_0)$. [9]
- $R|A = 1, Y = 0$ follows $Uniform(0, b_1)$.
- $R|A = 0, Y = 1$ follows $Uniform(c_0, 1)$.
- $R|A = 1, Y = 1$ follows $Uniform(c_1, 1)$.
- $c_0 < b_0 < 1$ and $c_1 < b_1 < 1$.

Since $b_0 < 1$ and $b_1 < 1$, $\Pr\{\tilde{Y} = 1\} = 1$ if $\tau_0 \in [b_0, 1]$ and $\tau_1 \in [b_1, 1]$. Therefore, the optimal thresholds satisfies the following,

$$p_{0,1} \cdot (1 - F_{R|0,1}(\tau_0)) = p_{1,1} \cdot (1 - F_{R|1,1}(\tau_1)), \qquad (21)$$

$$p_{0,1}\left(\frac{1 - \tau_0}{1 - c_0}\right) = p_{1,1}\left(\frac{1 - \tau_1}{1 - c_1}\right). \qquad (22)$$

For instance, if $p_{0,1} \cdot \frac{1-b_0}{1-c_0} \leq p_{1,1} \cdot \frac{1-b_1}{1-c_1}$, then $\tau_0 = b_0$ and $\tau_1 = 1 - \frac{p_{0,1}}{p_{1,1}}\frac{1-c_1}{1-c_0}$ are the solution to (13).

**Case 2**: consider the following scenario,

- $R|A = 0, Y = 0$ follows $Uniform(b_0, 1)$.
- $R|A = 1, Y = 0$ follows $Uniform(b_1, 1)$.
- $R|A = 0, Y = 1$ follows $Uniform(c_0, 1)$.
- $R|A = 1, Y = 1$ follows $Uniform(c_1, 1)$.
- $b_0 < c_0 < 1$ and $b_1 < c_1 < 1$.

Without loss of generality, assume that $p_{0,1} \leq p_{1,1}$. Using (21), we can see that $\tau_1$ and $\tau_0$ satisfy the following,

$$p_{0,1}\frac{1 - \tau_0}{1 - c_0} = p_{1,1} \cdot \frac{1 - \tau_1}{1 - c_1} \implies$$

$$\frac{1 - \tau_1}{1 - \tau_0} = \frac{p_{0,1}}{p_{1,1}}\frac{1 - c_1}{1 - c_0}, \tau_0 \in [c_0, 1], \ \tau_1 \in [c_1, 1]. \qquad (23)$$

Moreover, we can write $\Pr\{\tilde{Y} = 1\}$ as follows,

$$\Pr\{\tilde{Y} = 1\} = \frac{p_{0,1} \cdot \left(\frac{1-\tau_0}{1-c_0}\right) + p_{1,1} \cdot \frac{1-\tau_1}{1-c_1}}{p_{0,1} \cdot \frac{1-\tau_0}{1-c_0} + p_{0,0} \cdot \frac{1-\tau_0}{1-b_0} + p_{1,0} \cdot \frac{1-\tau_1}{1-b_1} + p_{1,1} \cdot \frac{1-\tau_1}{1-c_1}} \implies$$

$$\Pr\{\tilde{Y} = 1\} = \frac{p_{0,1} \cdot \frac{1}{1-c_0} + p_{1,1} \cdot \frac{1-\tau_1}{1-\tau_0}\frac{1}{1-c_1}}{p_{0,1} \cdot \frac{1}{1-c_0} + p_{0,0} \cdot \frac{1}{1-b_0} + p_{1,0} \cdot \frac{1-\tau_1}{1-\tau_0}\frac{1}{1-b_1} + p_{1,1} \cdot \frac{1-\tau_1}{1-\tau_0}\frac{1}{1-c_1}}$$

Since $\Pr\{\tilde{Y} = 1\}$ is a function of $\frac{1-\tau_1}{1-\tau_0}$, $\tau_0$ and $\tau_1$ are optimal if they satisfy (23). For instance, $\tau_0 = c_0$ and $\tau_1 = 1 - \frac{p_{0,1}}{p_{1,1}}(1 - c_1)$ are optimal thresholds.

### A.4 Restating Theorem 5 for the statistical parity (SP) fairness notion

Here we restate Theorem 5 for the statistical parity. The proof is similar to the proof of Theorem 5.

**Theorem 6.** *Let $Z$ be the predictor derived by thresholding $R \in \{\rho_1, \ldots, \rho_{n'}\}$ using thresholds $\tau_0, \tau_1$. Moreover, assume that accuracy (i.e., $\Pr\{Y = 1|Z = 1\}$) is increasing in $\tau_0$ and $\tau_1$, and $\Pr\{R = \rho_{n'}|A = a\} \leq \gamma, \forall a\{0, 1\}$. Then, under $\gamma$-SP (i.e., $|\Pr\{Z = 1|A = 0\} - \Pr\{Z = 1|A = 1\}| \leq \gamma$), one of the following pairs of thresholds is fair optimal.*

- $\tau_0 > \rho_{n'}$ *and* $\tau_1 = \rho_{n'}$ *(in this case, $\Pr\{R \geq \tau_0|A = 0\} = 0$).*

- $\tau_1 > \rho_{n'}$ *and* $\tau_0 = \rho_{n'}$ *(in this case, $\Pr\{R \geq \tau_1|A = 1\} = 0$).*

---

[9]$Uniform(p, q)$ denotes the uniform distribution in interval $[p, q]$.

## A.5 Deriving $\Pr\{\tilde{Y} = 1\}$ in terms of $\tilde{\tau}_0$ and $\tilde{\tau}_1$

Note that $(X, Y)$ and $\tilde{A}$ are conditionally independent given $A$.

$$\Pr\{\tilde{Y} = 1\} = \frac{\Pr\{\tilde{Z} = 1, Y = 1\}}{\Pr\{\tilde{Z} = 1\}} = \frac{\sum_{a,\tilde{a}\in\{0,1\}} \Pr\{r(X, \tilde{A}) \geq \tilde{\tau}_{\tilde{a}}, Y = 1, A = a, \tilde{A} = \tilde{a}\}}{\sum_{a,\tilde{a}\in\{0,1\}} \Pr\{r(X, \tilde{A}) \geq \tilde{\tau}_{\tilde{a}}, A = a, \tilde{A} = \tilde{a}\}}.$$

$$\Pr\{r(X, \tilde{A}) \geq \tilde{\tau}_a, Y = 1, A = a, \tilde{A} = a\} = \Pr\{A = a\} \cdot \Pr\{r(X, a) \geq \tilde{\tau}_a, Y = 1, \tilde{A} = a | A = a\} =$$

$$\Pr\{A = a\} \cdot \Pr\{r(X, a) \geq \tilde{\tau}_a, Y = 1 | A = a\} \cdot \Pr\{\tilde{A} = a | A = a\} = \frac{e^\epsilon}{1 + e^\epsilon} \overline{F}_{r(X,a),A,Y}(\tilde{\tau}_a, a, 1).$$

Similarly, we have,

$$\Pr\{r(X, \tilde{A}) \geq \tilde{\tau}_a, Y = 1, A = 1 - a, \tilde{A} = a\} = \frac{1}{1 + e^\epsilon} \overline{F}_{r(X,a),A,Y}(\tilde{\tau}_a, 1 - a, 1).$$

$$\Pr\{r(X, \tilde{A}) \geq \tilde{\tau}_a, A = a, \tilde{A} = a\} = \frac{e^\epsilon}{1 + e^\epsilon} \overline{F}_{r(X,a),A}(\tilde{\tau}_a, a).$$

$$\Pr\{r(X, \tilde{A}) \geq \tilde{\tau}_a, A = 1 - a, \tilde{A} = a\} = \frac{1}{1 + e^\epsilon} \overline{F}_{r(X,a),A}(\tilde{\tau}_a, 1 - a).$$

As a result,

$$\Pr\{\tilde{Y} = 1\} = \frac{e^\epsilon \sum_a \overline{F}_{r(X,a),A,Y}(\tilde{\tau}_a, a, 1) + \sum_a \overline{F}_{r(X,a),A,Y}(\tilde{\tau}_a, 1 - a, 1)}{e^\epsilon \sum_a \overline{F}_{r(X,a),A}(\tilde{\tau}_a, a) + \sum_a \overline{F}_{r(X,a),A}(\tilde{\tau}_a, 1 - a)}. \tag{24}$$

## A.6 Deriving equal opportunity fairness constraint in terms of $\beta_{\tilde{a},\hat{y}}$

For the equal opportunity, we have,

$$\Pr\{Z = 1 | Y = 1, A = 0\} = \Pr\{Z = 1 | Y = 1, A = 1\}.$$

$$\Pr\{Z = 1 | Y = 1, A = 0\} = \sum_{\tilde{a},\hat{y}} \Pr\{Z = 1 | Y = 1, A = 0, \tilde{A} = \tilde{a}, r(Z, \tilde{A}) = \hat{y}\} \Pr\{\tilde{A} = \tilde{a}, r(Z, \tilde{A}) = \hat{y} | Y = 1, A = 0\}$$

$$= \sum_{\tilde{a},\hat{y}} \Pr\{Z = 1 | Y = 1, A = 0, \tilde{A} = \tilde{a}, r(Z, \tilde{A}) = \hat{y}\} \Pr\{\tilde{A} = \tilde{a} | A = 0\} \Pr\{r(X, \tilde{a}) = \hat{y} | Y = 1, A = 0\}$$

$$= \sum_{\hat{y}} \beta_{0,\hat{y}} \frac{e^\epsilon}{1 + e^\epsilon} \Pr\{r(X, 0) = \hat{y} | Y = 1, A = 0\} + \sum_{\hat{y}} \beta_{1,\hat{y}} \frac{1}{1 + e^\epsilon} \Pr\{r(X, 1) = \hat{y} | Y = 1, A = 0\}$$

Similarly, we have,

$$\Pr\{Z = 1 | Y = 1, A = 1\} = \sum_{\tilde{a},\hat{y}} \Pr\{Z = 1 | Y = 1, A = 1, \tilde{A} = \tilde{a}, r(Z, \tilde{A}) = \hat{y}\} \Pr\{\tilde{A} = \tilde{a}, r(Z, \tilde{A}) = \hat{y} | Y = 1, A = 1\}$$

$$= \sum_{\tilde{a},\hat{y}} \Pr\{Z = 1 | Y = 1, A = 1, \tilde{A} = \tilde{a}, r(Z, \tilde{A}) = \hat{y}\} \Pr\{\tilde{A} = \tilde{a} | A = 1\} \Pr\{r(X, \tilde{a}) = \hat{y} | Y = 1, A = 1\}$$

$$= \sum_{\hat{y}} \beta_{0,\hat{y}} \frac{1}{1 + e^\epsilon} \Pr\{r(X, 0) = \hat{y} | Y = 1, A = 1\} + \sum_{\hat{y}} \beta_{1,\hat{y}} \frac{e^\epsilon}{1 + e^\epsilon} \Pr\{r(X, 1) = \hat{y} | Y = 1, A = 1\}$$

Therefore, the equal opportunity fairness notion can be written as follows,

$$\sum_{\hat{y}} \beta_{0,\hat{y}} \frac{1}{1 + e^\epsilon} \Pr\{r(X, 0) = \hat{y} | Y = 1, A = 1\} + \sum_{\hat{y}} \beta_{1,\hat{y}} \frac{e^\epsilon}{1 + e^\epsilon} \Pr\{r(X, 1) = \hat{y} | Y = 1, A = 1\}$$

$$= \sum_{\hat{y}} \beta_{1,\hat{y}} \frac{1}{1 + e^\epsilon} \Pr\{r(X, 1) = \hat{y} | Y = 1, A = 0\} + \sum_{\hat{y}} \beta_{0,\hat{y}} \frac{e^\epsilon}{1 + e^\epsilon} \Pr\{r(X, 0) = \hat{y} | Y = 1, A = 0\}$$

## A.7 Numerical Experiment

We compared EO and ES fairness notions in Table 2 after adding the following constraints to (13).

$$\Pr\{\text{No one is selected in 100 time steps}\} = \big(\Pr\{R < \tau_A\}\big)^{100}$$

$$= \big(\Pr\{A = 0\} \Pr\{R < \tau_0 | A = 0\} + \Pr\{A = 1\} \Pr\{R < \tau_1 | A = 1\}\big)^{100} \leq \psi, \tag{25}$$

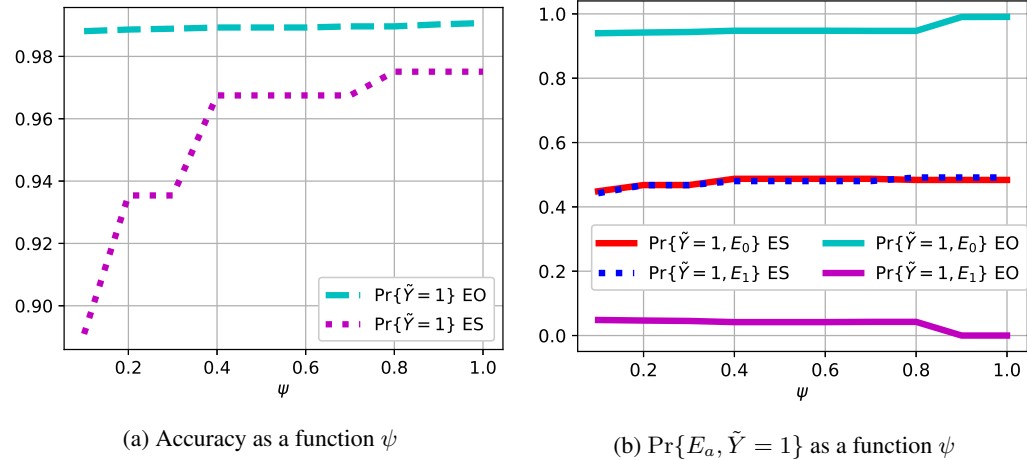

(a) Accuracy as a function $\psi$

(b) $\Pr\{E_a, \tilde{Y} = 1\}$ as a function $\psi$

Figure 2: Effect of the time constraint on the accuracy and disparity.

where $\psi$ was equal to 0.5 in that experiment. The above condition makes the experiment more practical and implies that there is a time limit in selecting an individual/applicant.

In this part, we study the effect of probability $\psi$ on the selection rate and accuracy. Smaller $\psi$ implies that the time constraint is more strict. Figure 2a illustrates the accuracy as a function of $\psi$. As expected, smaller $\psi$ worsens the accuracy under EO and ES. However, the parameter $\psi$ has relatively stronger impact on the accuracy under ES.

Figure 2b illustrates $\Pr\{E_a, \tilde{Y} = 1\}$ as a function of $\psi$. It shows that smaller $\psi$ is able to improve the disparity (i.e., $\gamma = |\Pr\{\tilde{Y} = 1, E_0\} - \Pr\{\tilde{Y} = 1, E_1\}|$) under EO.

## A.8 Proofs

*Theorem1.*

$$
\begin{aligned}
\Pr\{E_a, \tilde{Y} = 1\} &= \sum_{i=1}^{\infty} \Pr\{A_i = a, R_i = 1, Y_i = 1\} \times \Pr\{R_{i-1} = 0\} \times \cdots \times \Pr\{R_1 = 0\} \\
&= \frac{\Pr\{A = a, R = 1, Y = 1\}}{1 - \Pr\{R = 0\}}
\end{aligned}
$$

To ensure that $\Pr\{E_0, \tilde{Y} = 1\} = \Pr\{E_1, \tilde{Y} = 1\}$, we must have,

$$
\begin{aligned}
& \Pr\{E_0, \tilde{Y} = 1\} = \Pr\{E_1, \tilde{Y} = 1\} \\
\Leftrightarrow \quad & \frac{\Pr\{A = 0, R = 1, Y = 1\}}{1 - \Pr\{R = 0\}} = \frac{\Pr\{A = 1, R = 1, Y = 1\}}{1 - \Pr\{R = 0\}} \\
\Leftrightarrow \quad & \Pr\{A = 0, R = 1, Y = 1\} = \Pr\{A = 1, R = 1, Y = 1\}.
\end{aligned}
$$

$\square$

*Corollary 1.* Suppose the binary classifier $r(\cdot, \cdot)$ satisfies equal opportunity fairness, i.e., $R$ satisfies the following constraint,

$$
\Pr\{R = 1|A = 0, Y = 1\} = \Pr\{R = 1|A = 1, Y = 1\}. \tag{26}
$$

Based on Theorem 1, equation (26) implies equation (3) if and only if

$$
\Pr\{A = 0, Y = 1\} = \Pr\{A = 1, Y = 1\}.
$$

$\square$

*Theorem 2.* Let $\hat{\alpha}_{a,\hat{y}}$ be the solution to (5) and $\sum_{\hat{y},a} := \sum_{\hat{y}\in\{0,1\},a\in\{0,1\}}$. Note that $\hat{\alpha}_{a,\hat{y}}$ is also a feasible point for (4). We prove the theorem by contradiction. Assume that $\hat{\alpha}_{a,\hat{y}}$ is not the optimal solution for (4). Therefore, there exists $\overline{\alpha}_{a,\hat{y}}, a \in \{0,1\}, \hat{y} \in \{0,1\}$ such that,

$$\frac{\sum_{\hat{y},a} \overline{\alpha}_{a,\hat{y}} \cdot \Pr\{R=\hat{y}, Y=1, A=a\}}{\sum_{\hat{y},a} \overline{\alpha}_{a,\hat{y}} \cdot \Pr\{A=a, R=\hat{y}\}} > \frac{\sum_{\hat{y},a} \hat{\alpha}_{a,\hat{y}} \cdot \Pr\{R=\hat{y}, Y=1, A=a\}}{\sum_{\hat{y},a} \hat{\alpha}_{a,\hat{y}} \cdot \Pr\{A=a, R=\hat{y}\}}.$$

Let $\alpha_{\max} = \max_{a,\hat{y}} \overline{\alpha}_{a,\hat{y}}$. We define $\tilde{\alpha}_{a,\hat{y}}$ as follows,

$$\tilde{\alpha}_{a,\hat{y}} = \frac{\overline{\alpha}_{a,\hat{y}}}{\alpha_{\max}} \frac{\min_{\hat{y},a} \Pr\{A=a, R=\hat{y}\}}{\sum_{\hat{y},a} \frac{\overline{\alpha}_{a,\hat{y}}}{\alpha_{\max}} \cdot \Pr\{A=a, R=\hat{y}\}}. \tag{27}$$

It is easy to see that $\tilde{\alpha}_{a,\hat{y}} \in [0,1], a \in \{0,1\}, \hat{y} \in \{0,1\}$, and it is a feasible point for optimization problem (5). Moreover, we have,

$$\frac{\sum_{\hat{y},a} \overline{\alpha}_{a,\hat{y}} \cdot \Pr\{R=\hat{y}, Y=1, A=a\}}{\sum_{\hat{y},a} \overline{\alpha}_{a,\hat{y}} \cdot \Pr\{A=a, R=\hat{y}\}} = \frac{\sum_{\hat{y},a} \tilde{\alpha}_{a,\hat{y}} \cdot \Pr\{R=\hat{y}, Y=1, A=a\}}{\sum_{\hat{y},a} \tilde{\alpha}_{a,\hat{y}} \cdot \Pr\{A=a, R=\hat{y}\}}$$

$$> \frac{\sum_{\hat{y},a} \hat{\alpha}_{a,\hat{y}} \cdot \Pr\{R=\hat{y}, Y=1, A=a\}}{\sum_{\hat{y},a} \hat{\alpha}_{a,\hat{y}} \cdot \Pr\{A=a, R=\hat{y}\}}.$$

Note that since $\tilde{\alpha}_{a,\tilde{y}}$ and $\tilde{\alpha}_{a,\tilde{y}}, a \in \{0,1\}, \hat{y} \in \{0,1\}$ are feasible for problem (5), we conclude that,

$$\sum_{\hat{y},a} \tilde{\alpha}_{a,\hat{y}} \cdot \Pr\{R=\hat{y}, Y=1, A=a\} > \sum_{\hat{y},a} \hat{\alpha}_{a,\hat{y}} \cdot \Pr\{R=\hat{y}, Y=1, A=a\}.$$

This implies that $\hat{\alpha}_{a,\hat{y}}, a \in \{0,1\}, \hat{y} \in \{0,1\}$ is not optimal for (5). This is a contradiction, and $\hat{\alpha}_{a,\hat{y}}, a \in \{0,1\}, \hat{y} \in \{0,1\}$ is the solution for both (4) and (5).

Next, we prove the second part of the theorem.

Proof by contradiction. Assume that optimization problem (5) does not have a solution, and $\overline{\alpha}_{a,\hat{y}} \in [0,1], a \in \{0,1\}, \hat{y} \in \{0,1\}$ is the solution to (4). It is easy to see that $\tilde{\alpha}_{a,\hat{y}}$ define in (27) is a feasible point for (5). This is a contradiction because a linear optimization with a closed and bounded and non-empty feasible set has an optimal solution. This concludes the proof. $\qquad\square$

*Theorem 3.* Let predictor $Z$ be the solution to optimization problem (4). Without loss of generality, suppose $\Pr\{R=1, Y=1, A=0\} \leq \Pr\{R=1, Y=1, A=1\}$. Consider the following feasible point of optimization problem (4):

$$\hat{\alpha}_{00} = 0, \hat{\alpha}_{10} = 0, \hat{\alpha}_{01} = 1, \hat{\alpha}_{11} = \frac{\Pr\{R=1, Y=1, A=0\}}{\Pr\{R=1, Y=1, A=1\}}.$$

Using these parameters, we can derive a new predictor noted as $\hat{Z}$. Note that $\Pr\{R=1|\hat{Z}=1\} = 1$ holds. Moreover, Since $\hat{Z}$ is a suboptimal solution to optimization problem (4), $\Pr\{Y=1|\hat{Z}=1\} \leq \Pr\{Y=1|Z=1\}$ and we have,

$$|\Pr\{Y=1|Z^*=1\} - \Pr\{Y=1|Z=1\}| \leq |\Pr\{Y=1|Z^*=1\} - \Pr\{Y=1|\hat{Z}=1\}| \leq$$

$$|\Pr\{Y=1|Z^*=1\} - \Pr\{Y=1|R=1\}| + |\Pr\{Y=1|\hat{Z}=1\} - \Pr\{Y=1|R=1\}| \leq$$

$$\epsilon + |\Pr\{Y=1|\hat{Z}=1\} - \Pr\{Y=1|R=1\}|.$$

Moreover, we have,

$$\Pr\{Y=1|\hat{Z}=1\} = \Pr\{Y=1|\hat{Z}=1, R=1\} \cdot \underbrace{\Pr\{R=1|\hat{Z}=1\}}_{1} = \frac{\Pr\{Y=1|R=1\}}{\Pr\{\hat{Z}=1|R=1\}} \Pr\{\hat{Z}=1|Y=1, R=1\},$$

$$\Pr\{\hat{Z}=1|R=1\} = P(A=0|R=1) + P(A=1|R=1)\hat{\alpha}_{11},$$

$$\Pr\{\hat{Z} = 1 | Y = 1, R = 1\} = \Pr\{A = 0 | Y = 1, R = 1\} + \Pr\{A = 1 | Y = 1, R = 1\}\hat{\alpha}_{11},$$

$$\implies |\Pr\{Y = 1 | Z^* = 1\} - \Pr\{Y = 1 | Z = 1\}| \le$$
$$\epsilon + |\Pr\{Y = 1 | R = 1\} \, (1 - \underbrace{\frac{\Pr\{A = 0 | Y = 1, R = 1\} + \Pr\{A = 1 | Y = 1, R = 1\}\hat{\alpha}_{11}}{P(A = 0 | R = 1) + P(A = 1 | R = 1)\hat{\alpha}_{11}}}_{0}) |.$$

Since we assumed $\Pr\{A = a | Y = 1, R = 1\} \in \Pr\{A = a | R = 1\}, \forall a\{0,1\}$, the second term in above equation is zero, and the theorem has been proved.

$$\implies |\Pr\{Y = 1 | Z^* = 1\} - \Pr\{Y = 1 | Z = 1\}| \le \epsilon.$$

$\square$

*Lemma 1.* Based on the Theorem 1, predictor $Z$ satisfies the ES fairness notion if
$$\Pr\{Z = 1, A = 0, Y = 1\} = \Pr\{Z = 1, A = 1, Y = 1\}.$$

We use the law of total probability to rewrite the above condition using $\beta_{a,\hat{y}}$. Note that $\tilde{A}$ is derived directly from $A$ and is conditionally independent of $(X, Y)$ given $A$. We have,

$$\Pr\{Z = 1, A = 1, Y = 1\}$$
$$= \sum_{\tilde{a},\hat{y}} \Pr\{Z = 1, \tilde{A} = \tilde{a}, r(X, \tilde{A}) = \hat{y}, A = 1, Y = 1\}$$
$$= \sum_{\tilde{a},\hat{y}} \Pr\{Z = 1 | \tilde{A} = \tilde{a}, r(X, \tilde{A}) = \hat{y}, A = 1, Y = 1\} \cdot \Pr\{\tilde{A} = \tilde{a}, r(X, \tilde{A}) = \hat{y}, A = 1, Y = 1\}$$
$$= \sum_{\tilde{a},\hat{y}} \Pr\{Z = 1 | \tilde{A} = \tilde{a}, r(X, \tilde{A}) = \hat{y}\} \cdot \Pr\{\tilde{A} = \tilde{a}, r(X, \tilde{a}) = \hat{y} | A = 1, Y = 1\} \Pr\{A = 1, Y = 1\}$$
$$= \sum_{\tilde{a},\hat{y}} \beta_{\tilde{a},\hat{y}} \cdot \Pr\{\tilde{A} = \tilde{a} | A = 1, Y = 1\} \Pr\{r(X, \tilde{a}) = \hat{y} | A = 1, Y = 1\} \Pr\{A = 1, Y = 1\}$$
$$= \sum_{\tilde{a},\hat{y}} \beta_{\tilde{a},\hat{y}} \cdot \frac{\exp\{\epsilon \cdot \mathbb{I}(\tilde{a} = 1)\}}{1 + \exp\{\epsilon\}} \Pr\{r(X, \tilde{a}) = \hat{y}, A = 1, Y = 1\},$$

where indicator function $\mathbb{I}(s) = 1$ if statement $s$ is true, otherwise $\mathbb{I}(s) = 0$. Similarly, we have,

$$\Pr\{Z = 1, A = 0, Y = 1\} = \sum_{\tilde{a},\hat{y}} \beta_{\tilde{a},\hat{y}} \cdot \frac{\exp\{\epsilon \cdot \mathbb{I}(\tilde{a} = 0)\}}{1 + \exp\{\epsilon\}} \Pr\{r(X, \tilde{a}) = \hat{y}, A = 0, Y = 1\}.$$

Therefore, $Z$ satisfies the ES fairness notion if the following holds,

$$\Pr\{Z = 1, A = 0, Y = 1\} = \Pr\{Z = 1, A = 1, Y = 1\}$$
$$\Leftrightarrow \quad \beta_{0,0} \cdot e^{\epsilon} \cdot \Pr\{r(X, 0) = 0, A = 0, Y = 1\} + \beta_{0,1} \cdot e^{\epsilon} \cdot \Pr\{r(X, 0) = 1, A = 0, Y = 1\}$$
$$+ \beta_{1,0} \cdot \Pr\{r(X, 1) = 0, A = 0, Y = 1\} + \beta_{1,1} \cdot \Pr\{r(X, 1) = 1, A = 0, Y = 1\}$$
$$= \quad \beta_{0,0} \cdot \Pr\{r(X, 0) = 0, A = 1, Y = 1\} + \beta_{0,1} \cdot \Pr\{r(X, 0) = 1, A = 1, Y = 1\}$$
$$+ \beta_{1,0} \cdot e^{\epsilon} \Pr\{r(X, 1) = 0, A = 1, Y = 1\} + \beta_{1,1} \cdot e^{\epsilon} \cdot \Pr\{r(X, 1) = 1, A = 1, Y = 1\}.$$
$$(28)$$

$\square$

*Lemma 2.* We rewrite (28) as follows,

$$0 = \beta_{0,0} \cdot (e^{\epsilon} \cdot \Pr\{r(X, 0) = 0, A = 0, Y = 1\} - \Pr\{r(X, 0) = 0, A = 1, Y = 1\})$$
$$+ \beta_{0,1} \cdot (e^{\epsilon} \cdot \Pr\{r(X, 0) = 1, A = 0, Y = 1\} - \Pr\{r(X, 0) = 1, A = 1, Y = 1\})$$
$$+ \beta_{1,0} \cdot (\Pr\{r(X, 1) = 0, A = 0, Y = 1\} - e^{\epsilon} \Pr\{r(X, 1) = 0, A = 1, Y = 1\})$$
$$+ \beta_{1,1} \cdot (\Pr\{r(X, 1) = 1, A = 0, Y = 1\} - e^{\epsilon} \Pr\{r(X, 1) = 1, A = 1, Y = 1\}).$$

If Equation (8) holds, then $e^\epsilon \cdot \Pr\{r(X,a) = 1, A = a, Y = 1\} > 1, \forall a \in \{0,1\}$. Therefore, $(e^\epsilon \cdot \Pr\{r(X,0) = 0, A = 0, Y = 1\} - \Pr\{r(X,0) = 0, A = 1, Y = 1\})$ is a positive coefficient and $(\Pr\{r(X,1) = 1, A = 0, Y = 1\} - e^\epsilon \Pr\{r(X,1) = 1, A = 1, Y = 1\})$ is a negative coefficient. Therefore, above linear equation has a feasible point other than $\beta_{0,0} = \beta_{0,1} = \beta_{1,0} = \beta_{1,1} = 0$. $\quad\square$

*Theorem 4.* First, we find $\Pr\{\tilde{Y} = 1\}$ as a function of $\beta_{\tilde{a},\hat{y}}$. We have,

$$
\begin{aligned}
\Pr\{\tilde{Y} = 1\} &= \sum_{i=1}^{\infty} \Pr\{Z_i = 1, Y_i = 1\} \times \Pr\{Z_{i-1} = 0\} \times \cdots \times \Pr\{Z_1 = 0\} \\
&= \frac{\Pr\{Z = 1, Y = 1\}}{1 - \Pr\{Z = 0\}} = \frac{\Pr\{Z = 1, Y = 1\}}{\Pr\{Z = 1\}} = \Pr\{Y = 1 | Z = 1\},
\end{aligned}
$$

$$
\begin{aligned}
\Pr\{Z = 1, Y = 1\} &= \sum_{\tilde{a},\hat{y}} \Pr\{Z = 1, Y = 1 | \tilde{A} = \tilde{a}, r(X, \tilde{A}) = \hat{y}\} \cdot \Pr\{\tilde{A} = \tilde{a}, r(X, \tilde{A}) = \hat{y}\} \\
&= \sum_{\tilde{a},\hat{y}} \beta_{\tilde{a},\hat{y}} \cdot \Pr\{Y = 1 | \tilde{A} = \tilde{a}, r(X, \tilde{A}) = \hat{y}\} \cdot \Pr\{\tilde{A} = \tilde{a}, r(X, \tilde{A}) = \hat{y}\} \\
&= \sum_{\tilde{a},\hat{y}} \beta_{\tilde{a},\hat{y}} \cdot \Pr\{Y = 1, \tilde{A} = \tilde{a}, r(X, \tilde{A}) = \hat{y}\} \\
&= \sum_{\tilde{a},\hat{y}} \beta_{\tilde{a},\hat{y}} \cdot \Big( \Pr\{Y = 1, \tilde{A} = \tilde{a}, r(X, \tilde{A}) = \hat{y}, A = \tilde{a}\} \\
&\qquad\qquad + \Pr\{Y = 1, \tilde{A} = \tilde{a}, r(X, \tilde{A}) = \hat{y}, A = 1 - \tilde{a}\} \Big) \\
&= \sum_{\tilde{a},\hat{y}} \beta_{\tilde{a},\hat{y}} \cdot \Big( \Pr\{\tilde{A} = \tilde{a} | A = \tilde{a}\} \Pr\{Y = 1, r(X, \tilde{a}) = \hat{y} | A = \tilde{a}\} \Pr\{A = \tilde{a}\} \\
&\qquad\qquad + \Pr\{\tilde{A} = \tilde{a} | A = 1 - \tilde{a}\} \Pr\{Y = 1, r(X, \tilde{a}) = \hat{y} | A = 1 - \tilde{a}\} \Pr\{A = 1 - \tilde{a}\} \Big) \\
&= \sum_{\tilde{a},\hat{y}} \beta_{\tilde{a},\hat{y}} \cdot \Big( \frac{e^\epsilon}{1 + e^\epsilon} \Pr\{Y = 1, r(X, \tilde{a}) = \hat{y} | A = \tilde{a}\} \Pr\{A = \tilde{a}\} \\
&\qquad\qquad + \frac{1}{1 + e^\epsilon} \Pr\{Y = 1, r(X, \tilde{a}) = \hat{y} | A = 1 - \tilde{a}\} \Pr\{A = 1 - \tilde{a}\} \Big)
\end{aligned}
$$

$$
\begin{aligned}
\Pr\{Z = 1\} &= \sum_{\tilde{a},\hat{y}} \beta_{\tilde{a},\hat{y}} \cdot \Big( \frac{e^\epsilon}{1 + e^\epsilon} \Pr\{r(X, \tilde{a}) = \hat{y} | A = \tilde{a}\} \Pr\{A = \tilde{a}\} \\
&\qquad\qquad + \frac{1}{1 + e^\epsilon} \Pr\{r(X, \tilde{a}) = \hat{y} | A = 1 - \tilde{a}\} \Pr\{A = 1 - \tilde{a}\} \Big).
\end{aligned}
$$

We can see that $\Pr\{\tilde{Y} = 1\}$ is not a linear function in $\beta_{\tilde{a},\hat{y}}$. As a result, optimization problem (9) is not a linear program.

Now we prove the first part of the theorem using contradiction. Note that $\hat{\beta}_{\tilde{a},\hat{y}}$ is a feasible point for optimization problem (9). If $\hat{\beta}_{\tilde{a},\hat{y}}$ is not optimal for (9), then there exists $\overline{\beta}_{\tilde{a},\hat{y}}$ such that the objective function of (9) at $\overline{\beta}_{\tilde{a},\hat{y}}$ is higher than that at $\hat{\beta}_{\tilde{a},\hat{y}}$. With the similar approach as the proof of Theorem 1, we can show that the existence of $\overline{\beta}_{\tilde{a},\hat{y}}$ implies that $\hat{\beta}_{\tilde{a},\hat{y}}$ is not optimal for (10) because $\tilde{\beta}_{\tilde{a},\hat{y}}$ defined as follows is feasible and improves the objective function in (10).

$$
\tilde{\beta}_{\tilde{a},\hat{y}} = \frac{\overline{\beta}_{\tilde{a},\hat{y}}}{\beta_{\max}} \cdot \frac{\min_{\tilde{a},\hat{y}} P_A(\tilde{a}) \cdot e^\epsilon \cdot P_{r(X,\tilde{a})|A}(\hat{y}|\tilde{a}) + P_A(1 - \tilde{a}) \cdot P_{r(X,\tilde{a})|A}(\hat{y}|1 - \tilde{a})}{\sum_{\tilde{a},\hat{y}} \frac{\overline{\beta}_{\tilde{a},\hat{y}}}{\beta_{\max}} \Big( P_A(\tilde{a}) \cdot e^\epsilon \cdot P_{r(X,\tilde{a})|A}(\hat{y}|\tilde{a}) + P_A(1 - \tilde{a}) \cdot P_{r(X,\tilde{a})|A}(\hat{y}|1 - \tilde{a}) \Big)},
$$

where $\beta_{\max} = \max_{\tilde{a}\in\{0,1\},\hat{y}\in\{0,1\}} \overline{\beta}_{\tilde{a},\hat{y}}$ and $P_{r(X,\tilde{a})|A}(\hat{y}|\tilde{a}) := \Pr\{r(X, \tilde{a}) = \hat{y} | A = \tilde{a}\}$, $P_A(\tilde{a}) := \Pr\{A = \tilde{a}\}$ are defined to simplify notations.

This contradiction shows that $\hat{\beta}_{\tilde{a},\hat{y}}$ is optimal for optimization problem (9).

The proof of the second part of the theorem is similar to the first part. Proof by contradiction. Assume that $\overline{\beta}_{\tilde{a},\hat{y}}$ is the solution to optimization problem (9), and (10) does not have a solution. In this case,

we can show that $\tilde{\beta}_{\tilde{a},\hat{y}}$ defined as follows is a feasible point for (10).

$$\tilde{\beta}_{\tilde{a},\hat{y}} = \frac{\overline{\beta}_{\tilde{a},\hat{y}}}{\beta_{\max}} \cdot \frac{\min_{\tilde{a},\hat{y}} P_A(\tilde{a}) \cdot e^\epsilon \cdot P_{r(X,\tilde{a})|A}(\hat{y}|\tilde{a}) + P_A(1-\tilde{a}) \cdot P_{r(X,\tilde{a})|A}(\hat{y}|1-\tilde{a})}{\sum_{\tilde{a},\hat{y}} \frac{\overline{\beta}_{\tilde{a},\hat{y}}}{\beta_{\max}}\left(P_A(\tilde{a}) \cdot e^\epsilon \cdot P_{r(X,\tilde{a})|A}(\hat{y}|\tilde{a}) + P_A(1-\tilde{a}) \cdot P_{r(X,\tilde{a})|A}(\hat{y}|1-\tilde{a})\right)}.$$

Since (10) is a linear program and with a bounded and non-empty feasible set, it has at least a solution. This is a contradiction which proves the second part of the theorem. $\qquad\square$

*Theorem 5.* Note that the assumption that accuracy $\Pr\{Y=1|Z=1\}$ is increasing in $\tau_0$ and $\tau_1$ implies that an applicant with a higher score is more likely to be qualified and the policy with a larger threshold leads to higher accuracy. In other words, under this assumption, the decision-maker should make thresholds as large as possible to maximize the accuracy. We have,

$$\Pr\{Y=1|Z=1\} =$$
$$\Pr\{Y=1|R \geq \tau_0, A=0\}\Pr\{A=0|Z=1\} + \Pr\{Y=1|R \geq \tau_1, A=1\}\Pr\{A=1|Z=1\}.$$

Since $\Pr\{Y=1|Z=1\}$ is increasing in thresholds $\tau_0$ and $\tau_1$, it is optimal to select $\tau_0 = \rho_{n'}$ and $\tau_1 > \rho_{n'}$ if $\Pr\{Y=1|R=\rho_{n'}, A=0\} \geq \Pr\{Y=1|R=\rho_{n'}, A=1\}$. That is, no one with sensitive attribute $A=1$ is selected, and only an applicant with sensitive attribute $A=0$ and the highest possible score is selected. Note that $\tau_0$ and $\tau_1$ satisfy EO because we assumed that $\Pr\{R=\rho_{n'}|A=a, Y=1\} < \gamma$.

Similarly, it is optimal to select $\tau_1 = \rho_{n'}$ and $\tau_0 > \rho_{n'}$ if $\Pr\{Y=1|R=\rho_{n'}, A=1\} \geq \Pr\{Y=1|R=\rho_{n'}, A=0\}$. $\qquad\square$

*Lemma 3 .* Based on Theorem 1, to satisfy the ES fairness notion, the following should hold,

$$\Pr\{r(X,\tilde{A}) \geq \tilde{\tau}_{\tilde{A}}, A=0, Y=1\} = \Pr\{r(X,\tilde{A}) \geq \tilde{\tau}_{\tilde{A}}, A=1, Y=1\},$$

where

$$\begin{aligned}
&\Pr\{r(X,\tilde{A}) \geq \tilde{\tau}_{\tilde{A}}, A=a, Y=1\} \\
=\ & \Pr\{r(X,\tilde{A}) \geq \tilde{\tau}_{\tilde{A}}, A=a, Y=1, \tilde{A}=a\} + \Pr\{r(X,\tilde{A}) \geq \tilde{\tau}_{\tilde{A}}, A=a, Y=1, \tilde{A}=1-a\} \\
=\ & \Pr\{r(X,a) \geq \tilde{\tau}_a|Y=1, A=a\} \cdot \Pr\{\tilde{A}=a|Y=1, A=a\} \cdot \Pr\{Y=1, A=a\} \\
& + \Pr\{r(X,1-a) \geq \tilde{\tau}_{1-a}|Y=1, A=a\} \cdot \Pr\{\tilde{A}=1-a|Y=1, A=a\} \cdot \Pr\{Y=1, A=a\} \\
=\ & \Pr\{r(X,a) \geq \tilde{\tau}_a|Y=1, A=a\} \cdot \frac{e^\epsilon}{1+e^\epsilon} \cdot \Pr\{Y=1, A=a\} \\
& + \Pr\{r(X,1-a) \geq \tilde{\tau}_{1-a}|Y=1, A=a\} \cdot \frac{1}{1+e^\epsilon} \cdot \Pr\{Y=1, A=a\}
\end{aligned}$$

Therefore, the ES fairness notion is satisfied if and only if

$$\begin{aligned}
& \Pr\{r(X,0) \geq \tilde{\tau}_0|Y=1, A=0\} \cdot \frac{e^\epsilon}{1+e^\epsilon} \cdot \Pr\{Y=1, A=0\} \\
+\ & \Pr\{r(X,1) \geq \tilde{\tau}_1|Y=1, A=0\} \cdot \frac{1}{1+e^\epsilon} \cdot \Pr\{Y=1, A=0\} \\
=\ & \Pr\{r(X,1) \geq \tilde{\tau}_1|Y=1, A=1\} \cdot \frac{e^\epsilon}{1+e^\epsilon} \cdot \Pr\{Y=1, A=1\} \\
+\ & \Pr\{r(X,0) \geq \tilde{\tau}_0|Y=1, A=1\} \cdot \frac{1}{1+e^\epsilon} \cdot \Pr\{Y=1, A=1\}
\end{aligned}$$

$\qquad\square$