# OpenReview forum: "Fair Sequential Selection Using Supervised Learning Models"
_NeurIPS.cc/2021/Conference — NeurIPS 2021 Poster_

### Official Review · Reviewer_DxNT · 2021-07-14

**Rating:** 6
**Confidence:** 4

**Summary:**

This paper introduces a new task of making fair decisions in sequential selection problems. The authors have shown that existing fairness notions are not suitable for this problem and could lead to discrimination. Then the authors propose a new fairness notion called "Equalized Selection Rate (ESR)" for sequential selection problems and a post-processing approach is designed to satisfy the ESR constraint. The paper considers 2*2 different settings, i.e., whether the pre-trained model provides a binary decision or a continuous score, and whether the actual or noisy sensitive attribute is accessed. The paper also conducts experiments on FICO and Adult datasets.

**Limitations And Societal Impact:**

Yes

**Main Review:**

The paper studies a novel setting in fair machine learning. I can see that there are a lot of theoretical works done by this paper. However, I feel that this paper is not quite well-written as some key points are not explained or missing. In addition, how each section in the appendix is responding to each part in the main paper is not explicit (the authors should refer to the appendix sections in the main paper). These factors lower my score. My detailed comments are as follows.

1. Theorem 1 and A.1 need more explanations in the paper. The derivation of the first equation to Theorem 1 and the equation between Eq. (18) and (19) could show the key difference between the sequential selection setting and traditional supervised learning setting.

2. In the proof to Theorem 2, how the second constraint in Eq. (5) is used? I believe that it is hidden somewhere in the proof but the authors should explicitly show that.

3. The paper defines accuracy as Pr{\tide{Y}=1}. But to me this is just the true positive rate, not the accuracy, since it only considers the case where the individual is predicted positive. Could the authors clarify this?

4. The logic of Sections 4.1 and 3.1 are not consistent. In Section 3.1, the paper defines a probability to flit the decision of the pre-trained model. But in Section 4.1, the paper “trust” the score made by the pre-trained model and only aims to learn a threshold for making decisions.

I don’t have the expertise to check the differential privacy part of this paper.


**Time Spent Reviewing:**

3

---

> ### Author Response · Authors · 2021-08-10
> **Response to comments and addressing reviewer's concerns**
>
> $\textbf{Regarding the appendix}$.
>
> Thank you for pointing this out. We have already referred to the appendix in the main paper. However, we will be happy to improve the paper further. We hope that this does not negatively affect your score, because it is easily fixable.
> In the footnote of page 4, we have already referred to section A.1;
> In the footnote of page 5, we have already referred to section A.2;
> In line 256, we have already referred to section A.3;
> In line 271, we have already referred to section A.4;
> In the footnote of page 9, we have already referred to section A.5;
> In line 306, we have already referred to section A.6;
> Proofs are given in section A.7 (we will mention this in the paper).
>
> $\textbf{Comment:}$ Theorem 1 and A.1 need more explanations in the paper. The derivation of the first equation to Theorem 1 and the equation between Eq. (18) and (19) could show the key difference between the sequential selection setting and traditional supervised learning setting.
>
> $\textbf{Response:}$ Thank you for pointing out this. We will be happy to explain more about the first equation in the proof of theorem 1.
> Let $I_i=1$ be an event that individual $i$ at time slot $i$ is selected. Similarly, $I_i=0$ is an event that individual $i$ is not selected. Note that if an applicant is selected, the process stops.  We can write $\Pr\\{E_a,\tilde{Y}=1\\}$ as follows,
> $$\Pr\\{E_a,\tilde{Y}=1\\} = \Pr\\{I_1= 1,A_1=a,Y_1=1\\} + \Pr\\{I_1=0,I_2= 1,A_2=a,Y_2=1\\} + \Pr\\{I_1=0,I_2=0,I_3=1,A_3 =a,Y_3 = 1\\}+ ...  $$
> Since $R_i$ is a binary predictor, individual $i$ is selected if $R_i=1$. It implies that,
> $$ \Pr\\{E_a,\tilde{Y}=1\\}=  \Pr\\{R_1= 1,A_1=a,Y_1=1\\} + \Pr\\{R_1=0,R_2= 1,A_2=a,Y_2=1\\} + \Pr\\{R_1=0,R_2=0,R_3=1,A_3 =a,Y_3=1\\}+ .. $$
> Because $(R_i,A_i,Y_i)$ and $(R_j,A_j,Y_j)$ are i.i.d. random variables, we have,
> $$ \Pr\\{E_a,\tilde{Y}=1\\}=  \Pr\\{R_1= 1,A_1=a,Y_1=1\\} + \Pr\\{R_1=0\\}\Pr\\{R_2= 1,A_2=a,Y_2=1\\} + \Pr\\{R_1=0\\}\Pr\\{R_2=0\\}\Pr\\{R_3=1,A_3 =a,Y_3=1\\}+ .. $$
> Note that  $(R_i,A_i)$ and $(R_j,A_j)$ and $(R,A)$ have the same distribution. Therefore,
> $$ \Pr\\{E_a,\tilde{Y}=1\\}=  \Pr\\{R= 1,A=a,Y=1\\} + \Pr\\{R=0\\}\Pr\\{R= 1,A=a,Y=1\\} + \Pr\\{R=0\\}\Pr\\{R=0\\}\Pr\\{R_3=1,A_3 =a,Y=1\\}+ ... $$
> As you can see, the above equation is a geometric series. We have,
> $$ \Pr\\{E_a,\tilde{Y}=1\\}= \frac{\Pr\\{R= 1,A=a,Y=1\\}}{1-\Pr\\{R=0\\}}$$
>
> We can have a similar argument for the equation between Eq. (18) and (19).
>
> We will add this explanation to section A.1 and the proof of Theorem 1. We hope that the reviewer reconsiders his/her score because this issue is easily fixable.
>
> $\textbf{Comment:}$ In the proof to Theorem 2, how the second constraint in Eq. (5) is used? I believe that it is hidden somewhere in the proof but the authors should explicitly show that.
>
> $\textbf{Response:}$ Thank you for pointing this out. The second constraint in Eq. (5) has been used before line 557 to cancel out the denominators. We will improve the proof of theorem 2 in the final version.
>
> $\textbf{Comment:}$ The paper defines accuracy as $\Pr\\{\tilde{Y}=1\\}$. But to me this is just the true positive rate, not the accuracy, since it only considers the case where the individual is predicted positive. Could the authors clarify this?
>
>
> $\textbf{Response:}$ Thank you for pointing out this. Yes, we will clarify this in the final version. As we can see in the equation after line 189, $\Pr\\{\tilde{Y} =1 \\} = \Pr\\{Y=1|Z=1\\}$ which is not the true positive rate.
>
>
> $\textbf{Comment:}$  The logic of Sections 4.1 and 3.1 are not consistent. In Section 3.1, the paper defines a probability to flit the decision of the pre-trained model. But in Section 4.1, the paper “trust” the score made by the pre-trained model and only aims to learn a threshold for making decisions.
>
> $\textbf{Response:}$ We are sorry for the confusion. In both section 4.1 and section 3.1, the goal is to find a fair binary predictor.
> The pre-trained model in section 3.1, is a binary classifier. Therefore, we have to flit the decision to satisfy the ESR notion.  However, in section 4.1, the pre-trained model is not a classifier. It is a score function in [0,1].  In order to find a fair binary predictor using a score function, the easiest way is to use two thresholds to map the score function to a binary decision. This does not mean that we trust the pre-trained model. This is because we need a binary decision.
>
> Since you mentioned "trust", it is important to say that the quality of predictor $R$ affects the performance of binary decision $Z$. If $R$ is not an informative score function, $Z$ will not be an accurate predictor. More specifically, The following theorem identifies the condition under which we can trust predictor $R$. We will add this theorem to the final version.
>
> $\textsf{Theorem:}$ Let $Z^*$ be the optimal binary predictor that can be obtained using joint distribution of $(X,A,Y)$ through an in-processing method. Assume that  there exits $\underline{\tau}$ such that $\Pr\\{Y=1|Z^*=1\\} - \Pr\\{Y=1|R\geq \tau\\} \leq \epsilon, \forall \tau>\underline{\tau}$. Let $Z$ be the predictor obtained using the optimal thresholds of optimization problem (13). Then, we have,
> $$|\Pr\\{Y=1|Z^*=1\\} - \Pr\\{Y=1|Z=1\\}| \leq \epsilon$$
>
> $\textsf{Proof sketch:}$ Note that based on our definition, we can show that $\Pr\\{Y=1|Z=1\\}$ is the accuracy of binary predictor $Z$.
>
> We prove this theorem by contradiction. Assume that $|\Pr\\{Y=1|Z^*=1\\} - \Pr\\{Y=1|Z=1\\}| > \epsilon$.
>
> Conisider two thresholds $\tilde{\tau}_0$, $\tilde{\tau}_1$ such that $\tilde{\tau}_0 \geq \underline{\tau}$ and $\tilde{\tau}_1\geq \underline{\tau}$ and they are the feasible point of optimization problem (13) (wee can show that these two thresholds always exist). Let $\tilde{Z}$ be the predictor obtained by thresholding $R$ using $\tilde{\tau}_0$ and $\tilde{\tau}_1$.
> Since   $\tilde{\tau}_0 \geq \underline{\tau}$ and $\tilde{\tau}_1\geq \underline{\tau}$ and $\Pr\\{Y=1|Z^*=1\\} - \Pr\\{Y=1|R\geq \tau\\} \leq \epsilon, \forall \tau>\underline{\tau}$, we can show that $ \Pr\\{Y=1|\tilde{Z}=1\\} > \Pr\\{Y=1|Z=1\\}$. That is, $\tilde{Z}$ has a better accuracy compared to $Z$. This is a contradiction because $Z$ is derived from the optimal thresholds of optimization problem (13).
>
> This theorem implies that if $R$ is sufficiently informative about $Y=1$, then predictor $Z$ obtained by thresholding $R$ is sufficiently accurate. The theorem and the complete proof will be added to the paper.

---

> > ### Author Response · Authors · 2021-09-01
> > **Message to reviewer**
> >
> > We would like to thank the reviewer for the valuable comments. We hope our response has addressed all your concerns. Please let us know if you have any other questions, and we are happy to discuss more. If you’re satisfied with our response, we sincerely hope you could reconsider the rating.

---

### Official Review · Reviewer_hV6x · 2021-07-15

**Rating:** 6
**Confidence:** 4

**Summary:**

This paper studies how to "fairly" select a single candidate from a repeated i.i.d. classification perspective "selection process". They contribute a new fairness notion that is unique in the selection setting, however, it is not that different from previously contributed group fairness notions. They provide sound technical methods to post-process predictors to satisfy the fairness notion and consider selection in the private attribute setting.
However, the selection setting is not interesting enough without decisions at time step i affecting those at time step i+1, moreover, as the paper studies the  single selection setting, it is not that different from just classification.
The paper is novel and interesting in it's combination of all those elements, but there is no sufficient novelty in each direction alone to stand out. The paper is clearly written and easy to follow and technically sound. I think if the paper had a slightly more realistic selection setting (any sort of dynamics from one step to another), then it would be a clear accept. But without that, I think the paper is borderline.


**Limitations And Societal Impact:**

The papers introduces a fairness constraint on selection problems. Downstream impact of this fairness condition or it's ethical viability are not adequately discussed in the paper. Especially in selection problems, it is important to understand long term impact of a fairness notion rather than short term.

**Main Review:**

Originality: The paper considers a new fairness criterion for selection problems of finite resources. The fairness notion is similar to existing notions and has the same form of equality of group wise probabilities. Fairness in selection problems has previously been studied. Guaranteeing fairness when only privatized attributes are available has been studied in classification. The novelty in this paper is studying this new fairness notion in the selection setting (repeated i.i.d. classification) with private attributes (only at test time).  The postprocessing approach is a very interesting technical result and approach, and not obvious. However, the selection setting considered is only studied for a single selection, does not study dynamics of selections on each other (because it studies single selections) and only studies the setting for binary sensitive attributes (but I understand this may only complicate theory but not the approach).

Quality: The theorems and technical details are sound and complete. Some crucial experimental details are missing especially how 1) they simulated selection problems from the static datasets and 2) how the optimization was performed and thresholds chosen for EO and the baselines and 3) how exactly they computed the statistics. Furthermore, I am unsure about one of the results in table 1.

Clarity: The paper is clearly written and easy to follow. The basis of the paper is the following sentence  “decisions are fair if the (qualified) applicants from different groups are
selected at the same rate.”,  this implies in classification that P(R=1|Y=1,A=0)=P(R=1|Y=1,A=1) (EO) but Theorem 1 says otherwise, what the authors actually mean is that decisions are fair if we select the same number of qualified individuals from different groups and not “rates”. I think the proposed notion (ESR) is fundamentally very very different from EO as it is not a conditional statement .

Significance: I think this paper misses out an incredible opportunity to study selection problems by only focusing on the m=1 case and assumes i.i.d. selections and no impact of previous selections on the following ones. Due to this, I feel the "selection" aspect of the paper is not that novel and so it goes back to studying the ESR notion in classification problems and trying to argue for it (from theorem 1). However, the paper does not attempt to justify ESR in the classification  from an ethical perspective or long term perspective but only cites the EEOC.

Limitations:  1) studying binary attributes, 2) m=1 setting "single selection", 3) i.i.d. between selections which is not a reasonable assumption in reality, 4) not enough argumentation, especially from a societal perspective, for why one wants ESR instead of EO.

Comments:

 re footnote 1: College admission is a subset selection problem and not sequential selection, hiring can be argued as sequential selection.

How calibrated is the pretrained model?

ESR extension to m>1 is not obvious what it means, also authors don’t write in words what ESR signifies: The probability that we select a qualified A=0 individual vs A=1 individual should be the same, that is not the natural extension of EO.

Theorem 1 is illustrative but is obvious from the definition 1 and so is corollary 1.

Theorem 2 is nice, I think the proof needs to have all the steps written out (cancelling of terms more clearly) to be more easily readable.

I want to see a comparison or a theoretical attempt at the difference between post-processing and in-processing with fairness, something like Theorem 5.6 in Equality of Opportunity in Supervised Learning  (Hardt et al., 2016).

I think it is reasonable to remove Assumption 1 and assume one only observes the noisy attribute at test and train, if not I would like to see an impossibility result to justify assumption 1 as it is not reasonable to assume a discrepancy between train and test in observing the sensitive attribute.

In table 1, the result of the 0 probability of selecting the minority group is a bit odd. Consider the following  test: at each step, we sample a random individual from the population, the probability that this individual is Black and qualified is P(A=0,Y=1), if a classifier satisfies EO, then it cannot be the case that the probability of selecting a Black individual is 0 if we repeat this selection process. I need more details on how the testing was accomplished.

Constraint 17 is too artificial with the ½ threshold. I think it is okay to consider m=1 in theorems, but in experiments not showing m>1 is a big limitation.

What can make the paper really improve:
1) considering m>1
2) Considering that decisions at time step i, affect the distribution of individuals at time step i+1.
3) theoretical results to showcase the issue with EO shown in the experiments



---------------------------------------------

Post response: I have read the authors responses. The authors do acknowledge and address some of the limitations of their study, however, I remain on my previous score as the current problem studied is still limited by the lack of any dynamics governing the population or meaningful experimental study.

**Time Spent Reviewing:**

5

---

> ### Author Response · Authors · 2021-08-10
> **Response to comments and addressing reviewer's concerns**
>
> $\textbf{Originality}$
>
> $\textbf{A1. Regarding single selection:}$ As mentioned in the paper (line 128), since we are considering an i.i.d. setting, our results can be applied to a scenario where $m>1$. If $m>1$, then we have to repeat the process until $m$ applicants are selected.
> When $m>1$, then ESR implies that the probability that each position is filled with an applicant from group 0 is the same as that from group 1, $\Pr\\{\tilde{Y}=1\\}$ is the probability that a given position is filled by a qualified applicant.
>
> $\textbf{A2. Regarding dynamic setting:}$ Thank you for mentioning this. Yes, considering a dynamic setting would be an interesting extension and we are working on it as future work. Specifically, the participation and/or features of applicants in the future may be affected by the current decisions. For example, the high rejection rate may drive away applicants from certain groups; the applicants may exert more effort to improve features and increase the chance of being selected in the future. However, this dynamic is not the focus of the current paper. It's worth mentioning that in some applications, the current decision may not affect the feature vector of the future applicants if the decision-making process spans over a short time horizon. Consider an example of job applications, if the decision-maker rejects a person for not having a master’s degree, it is not easy for future applicants to change their qualifications by earning a master’s degree as it may take a few years and filing a vacant position takes only a few weeks.
>
> We totally agree with the reviewer that the dynamic setting is an important extension, and an i.i.d. setting applies to scenarios with a short time span. We hope that the reviewer does not penalize the paper because i.i.d. setting is also applicable to some real-world scenarios.
>
> $\textbf{A3. Binary sensitive attributes:}$ Thank you for mentioning this. Our model can be easily extended to settings with non-binary sensitive attributes, the qualitative results will remain valid. However, considering non-binary sensitive attributes only complicates the math and we do not gain anything in terms of technical contributions and qualitative results. We hope the reviewer does not penalize us for this.
>
> $\textbf{Quality}$
>
> $\textbf{B1. They simulated selection problems from the static datasets:}$ The only thing that we need for solving optimization problems (5) (10) (13) (15) is the joint distribution of $(R,A,Y)$. We can estimate this from the dataset. FICO dataset itself includes this distribution (Experiment 1). For experiment 2, we estimated the distribution using the Adult dataset. We will include this explanation in the final version of the paper.
>
> $\textbf{B2. How the optimization was performed and thresholds chosen for EO and the baselines:}$
> In experiment 1, FICO scores are discrete. As we mentioned in line 261, if the scores are discrete, we can easily find the optimal thresholds using an exhaustive search. Even if the fairness notion is EO, exhaustive search still works.
>
> In experiment 2, we are solving the optimization problem (10) under ESR or EO or no fairness constraint. Optimization (10) is linear in $\beta_{\tilde{a},\hat{y}}$ under ESR or no fairness constraint. In line 534 of the appendix, we showed that  EO can be written as a linear constraint in $\beta_{\tilde{a},\hat{y}}$. Therefore, the optimization problem (10) for EO and baseline were solved using the simplex method.
>
> $\textbf{B3. How exactly they computed the statistics.}$ In the first experiment, the statistics are included in the dataset. In particular, FICO score dataset includes $\Pr\\{R\leq \rho | A=a\\}$, $\Pr\\{Y=1|R=\rho, A=a\\}$ and $\Pr\\{A=a\\}$. With these three distributions, we can calculate the joint distribution of $(R,Y,A)$.
>
> In the second experiment, for finding the joint distribution of $(R,A,Y)$, we need to know 8 numbers/parameters ($\Pr\\{R=\rho,A=a,Y=y\\}, \rho \in \\{0,1\\}, a \in \\{0,1\\}, y \in \\{0,1\\} $) because $R$, $A$, $Y$ are binary. $\Pr\\{R=\rho,A=a,Y=y\\}$ can be estimated by counting and averaging.
>
> $\textbf{Clarity}$
>
> $\textbf{C1. Regarding ESR:}$ We agree with the reviewer. ESR is fundamentally different from EO and it tries to select the same number of qualified individuals from different groups. Specifically, the primary goal of ESR is to ensure the limited positions to be filled with a diverse group of people. Consider an example where 100 applicants compete for 3 positions, among all applicants 90 are from group 0 while 10 are from group 1. Under EO, it’s very likely that 3 positions are all filled with people in group 0 and zero from group 1, leading to a lack of diversity. As such, to improve the diversity when the number of selections is very limited we can use the (approximate) ESR notion. We will improve the argument about the ESR notion.
>
> $\textbf{Significance:}$
>
> $\textbf{D1. Regarding single selection and no dynamic:}$ We have already discussed these issues in the originality section. See parts A1 and A2.
>
> $\textbf{D2. Selection aspect is not novel:}$ It is worth mentioning that in a selection problem due to the competition, the objective function should be defined differently as compared to classification problems (as we did in this paper). Moreover, the post-processing method proposed in (Hardt et al. 2016) leads to a linear problem in classifications while leading to a non-convex optimization in selection problems; one of our contributions is that we proposed an approach to solving this non-convex optimization. Therefore, we believe that the selection aspect of the problem brings novelty to this paper.
>
> $\textbf{D3. Justification of the ESR notion.}$  As we mentioned in C1, the primary goal of ESR is to increase diversity. Note that the ESR notion resembles the Rooney Rule -- a policy widely used in the interview process and it implies that at least one candidate from the minority group should be invited for an interview when filling a vacant position. We will add more discussion on the ESR notion (see more justification in C1).
>
> $\textbf{Limitations:}$
>
> $\textbf{E1. 1) studying binary attributes}$, $\textbf{2) m=1 setting "single selection"}$, $\textbf{3) i.i.d. between selections which is not a reasonable assumption in reality}$,
>
> Please see detailed responses to the “Originality” (parts A1, A2, A3). Here is a summary: 1) Non-binary attributes do not change our results or methods, but only make the notions and math messy; 2) Our method works for $m>1$: we only need to repeat the selection process until $m>1$ applicants are selected; 3) i.i.d. selections are suitable for scenarios where the decision-making process spans over a short time horizon.
>
> $\textbf{E2. Not enough argumentation, especially from a societal perspective, for why one wants ESR instead of EO.}$
>
> ESR can effectively improve diversity when the number of selections is very limited. More justifications are provided in parts C1 and D3.
>
> $\textbf{comments:}$
>
> $\textbf{F1. re footnote 1: College admission is a subset selection problem and not sequential selection}$
>
> It is true that some admission processes can be modeled as a subset selection problem. However, some colleges indeed use rolling admissions to select students (please see https://www.princetonreview.com/college-advice/rolling-admission), which can be cast as a sequential selection problem.
>
> $\textbf{F2. How calibrated is the pre-trained model?}$
>
> We do not have any constraint on the pre-trained model. The method proposed in optimization problems (5) (10) (13) (15) works regardless of the calibration of the pre-trained model. However, the accuracy of the post-processing method is affected by the calibration/quality of the pre-trained model. Please see our response to the last comment of $\textbf{Reviewer hV6x}$ to see the theorem on quality/calibration of $R$.
>
> $\textbf{F3. ESR extension to m>1 is not obvious what it means, }$ $\textbf{also authors don’t write in words what ESR signifies:} $
> $\textbf{ The probability that we select a qualified A=0 individual vs A=1 individual should be the same,}$ $\textbf{ that is not the natural extension of EO.}$
>
> Regarding the extension to $m>1$: as mentioned above, to make multiple selections, we can repeat the selection process until $m>1$ applicants are selected. In this case, ESR implies that the probability that each vacant position is filled by a qualified applicant from group 0 is equal to the probability that it is filled by a qualified applicant from group 1.
>
> It is true that ESR is not an extension of EO as they are equalizing different measures. The primary goal of ESR is to improve diversity for scenarios where the number of positions is very limited. It’s similar to EO in the sense that both care about qualified individuals. We will clarify this in the paper.
>
>
> $\textbf {F4. Theorem 2 is nice, proof should be improved}$
>
> Sure. We will make it more readable. Thanks for the suggestion.
>
> $\textbf{F5. a comparison or a theoretical attempt at difference between post-processing and in-processing}$ $\textbf{ something like Theorem 5.6 in Equality of Opportunity in Supervised Learning (Hardt et al., 2016).}$
>
> Please refer to our response to the last comment of $\textbf{Reviewer hV6x}$ to see the theorem on quality/calibration of $R$.

---

> > ### Author Response · Authors · 2021-08-10
> > **(continued) Response to comments and addressing reviewer's concerns**
> >
> >
> > $\textbf{F6. I think it is reasonable to remove Assumption 1 and assume one only observes the noisy attribute at test and train,}$
> >
> > Yes, we can remove Assumption 1 and assume that the noisy sensitive attributes are observable during training. In this case, $\Pr\\{\tilde{A} = 0\\}$ is known and from which we can derive $\Pr\\{A=0\\}$. Specifically, according to the randomized response, we have,
> >
> > $$\Pr\\{\tilde{A} = 0 \\}=\Pr\\{\tilde{A} = 0|A=0 \\} \Pr\\{A=0\\}+\Pr\\{\tilde{A} = 0|A=1 \\} (1-\Pr\\{A=0\\})$$
> > $$=\frac{e^\epsilon}{1+e^\epsilon} \Pr\\{A=0\\}+\frac{1}{1+e^\epsilon} (1-\Pr\\{A=0\\})  $$
> > $$ \implies \Pr\\{A=0\\} = \frac{(1+e^{\epsilon}) \Pr\\{\tilde{A}=0\\}-1  }{e^{\epsilon}-1}$$
> >
> > Therefore, from the noisy sensitive attributes in the training data, we can first derive $\Pr\\{A=0\\}$ and then use the proposed methods. We will add this discussion to the paper.
> >
> >
> > $\textbf{F7. In table 1, the result of the 0 probability of selecting the minority group is a bit odd.}$ Consider the following test: at each step, we sample a random individual from the population, the probability that this individual is Black and qualified is P(A=0,Y=1), if a classifier satisfies EO, then it cannot be the case that the probability of selecting a Black individual is 0 if we repeat this selection process. I need more details on how the testing was accomplished.
> >
> > It can be justified as follows. In FICO score dataset, the white group is more qualified than the black, i.e., $\Pr\\{Y=1|A=0\\} > \Pr\\{Y=1|A=1\\}$. To select a qualified applicant, it is better for the decision-maker to select from the white group and meanwhile satisfying EO.
> > In this example, what happens is that $\Pr\\{R>\tau_a|Y=1,A=a\\}$ is too small for both groups, i.e., $\Pr\\{R>\tau_0|A=0,Y=1\\}= 0.000262703408197, \Pr\\{R>\tau_1|A=1,Y=1\\}= 0.0$. Even though 0.001-EO fairness is satisfied, we end up with selecting no one from the black group.
> >
> > $\textbf{F8. Constraint 17 is too artificial with the ½ threshold.}$ $\textbf{ I think it is okay to consider m=1 in theorems, but in experiments not showing m>1 is a big limitation.}$
> >
> >
> > We have conducted the experiment for different thresholds, the results can be found in Appendix A.6. For $m>1$, as mentioned earlier, we only need to repeat the selection process until $m$ applicants are selected. In this case, the accuracy would be the probability that a qualified applicant is selected for each position, and the fair predictor can still be found using optimization problems (5) (10) (13) (15). Because ESR implies that the probability that a given vacant position is filled by a qualified applicant from group 0 is equal to that from group 1, the numerical results remain the same even if $m>1$.
> >
> > $\textbf{What can make the paper really improve:}$
> >
> > $\textbf{F9. considering m>1: }$
> >
> > Please see parts A1 and F8.
> >
> >
> > $\textbf{F10. Considering that decisions at time step i, affect the distribution of individuals at time step i+1.}$
> >
> > Please see part A2.
> >
> > $\textbf{F11. theoretical results to showcase the issue with EO shown in the experiments}$
> >
> > Yes, we can prove the following theorem, which will be added to the paper:
> >
> > $\textsf{Theorem:}$ Let $Z$ be the predictor that is derived by thresholding $R$ using $\tau_0$ and $\tau_1$. Moreover, assume that $\Pr\\{Y=1|Z=1\\}$ is increasing in $\tau_0$ and $\tau_1$. Then, under $\gamma$-EO fairness notion, the maximum accuracy is obtained if $\Pr\\{R>\tau_0|A=0,Y=1\\} =0 $ or $\Pr\\{R>\tau_1|A=1,Y=1\\} =0 $.
> >
> > $\textsf{Proof Sketch:}$ Assume that $\Pr\\{R>\tau_0|A=0,Y=1\\} \neq 0 $ and $\Pr\\{R>\tau_1|A=1,Y=1\\} \neq 0 $ and |$\Pr\\{R>\tau_0|A=0,Y=1\\} -\Pr\\{R>\tau_1|A=1,Y=1\\}| \leq \gamma $, and $\tau_0$ and $\tau_1$ are the solution to (13) and maximizing the accuray. Since $\Pr\\{R>\tau_a|A=a,Y=1\\} \neq 0 $ and accuray is increasing in $\tau_0$ and $\tau_1$, we can show that by increasing at least one of $\tau_0$ or $\tau_1$ accuracy increases without violating $\gamma$-EO. This is contradiction. Therefore, $\Pr\\{R>\tau_0|A=0,Y=1\\} = 0 $ or $\Pr\\{R>\tau_1|A=1,Y=1\\} = 0 $.
> >
> > $\textbf{Limitations And Societal Impact:}$
> >
> > $\textbf{G1.}$ The papers introduces a fairness constraint on selection problems. Downstream impact of this fairness condition or it's ethical viability are not adequately discussed in the paper. Especially in selection problems, it is important to understand long term impact of a fairness notion rather than short term.
> >
> > We discussed this issue in part A2, which will be added to the paper.

---

> > > ### Comment · Reviewer_hV6x · 2021-08-23
> > > **Thank you for your response**
> > >
> > > Thank you for your very detailed response!
> > >
> > > Can you elaborate more on point F.11, can you elaborate on the proof a bit more please?
> > >
> > > I will write a response to all the other points soon.

---

> > > > ### Author Response · Authors · 2021-08-23
> > > > **Regarding the theorem**
> > > >
> > > > Thank you for your reply.
> > > >
> > > > Theorem in F.11 is stated for a case where $R$ is continuous in [0,1].
> > > >
> > > > In order to make it easier to understand, we would like to re-state our theorem for cases where  $R$ is discrete and $R\in \\{\rho_1,\ldots,\rho_{n’} \\}$. Note that in experiment 1, the scores are also discrete.
> > > >
> > > > $\textsf{Theorem:}$ Let $Z$ be the predictor derived by thresholding $R\in \\{\rho_1,\ldots,\rho_{n’} \\}$ using $\tau_0$ and $\tau_1$. Moreover, assume that accuracy (i.e., $\Pr\\{Y=1|Z=1\\}$) is increasing in $\tau_0$ and $\tau_1$ and $\Pr\\{R=\rho_{n’}|A=a,Y=1\\}<\gamma, \forall a$. Then, under $\gamma$-EO fairness notion, one of the following pairs of thresholds is fair optimal.
> > > > 1. $\tau_0 > \rho_{n’}$ and $\tau_1 =\rho_{n’} $ (in this case, $\Pr\\{R\geq \tau_0|A=0,Y=1\\} = 0$)
> > > > 2. $\tau_1 > \rho_{n’}$ and $\tau_0 =\rho_{n’} $ (in this case, $\Pr\\{R\geq \tau_1|A=1,Y=1\\} = 0$)
> > > >
> > > > $\textsf{Proof:}$
> > > > Note that the assumption that accuracy $\Pr\\{Y=1|Z=1\\}$ is increasing in $\tau_0$ and $\tau_1$ implies that an applicant with a higher score is more likely to be qualified and the policy with a larger threshold leads to higher accuracy.
> > > >
> > > > In other words, under this assumption, the decision-maker should make thresholds as large as possible to maximize the accuracy. Because we assumed that $\Pr\\{R=\rho_{n'}|A=a,Y=1\\}<\gamma$,  it is optimal to select $\tau_0 = \rho_{n'}$ and $\tau_1 > \rho_{n'}$ if $\Pr\\{Y=1|R=\rho_{n'} ,A=0\\} \geq \Pr\\{Y=1|R=\rho_{n'} ,A=1\\} $. That is, no one with sensitive attribute $A=1$ is selected, and only an applicant with sensitive attribute $A=0$ and the highest possible score is selected.
> > > >
> > > > Similarly, it is optimal  to select $\tau_1 = \rho_{n'}$ and $\tau_0 > \rho_{n'}$ if $\Pr\\{Y=1|R=\rho_{n'} ,A=1\\} \geq \Pr\\{Y=1|R=\rho_{n'} ,A=0\\} $.
> > > >
> > > > We are happy to include this in the paper. Please let us know if it is still confusing.

---

> > > > > ### Comment · Reviewer_hV6x · 2021-08-31
> > > > > **On the theorem**
> > > > >
> > > > > Thank you for the detailed proof and theorem.

---

> > > > > > ### Author Response · Authors · 2021-09-01
> > > > > > **Message to reviewer**
> > > > > >
> > > > > > Thanks for your reply! We hope our proof and theorem have addressed all your concerns. Since we are approaching the end of the discussion period, please let us know if you have any other questions, and we are happy to discuss more. If you’re satisfied with our response, we sincerely hope you could reconsider the rating.

---

### Official Review · Reviewer_ffFj · 2021-07-17

**Rating:** 5
**Confidence:** 4

**Summary:**

The paper proposes a notion of algorithmic fairness for the setting of sequential selection. Namely, a situation where candidates arrive one after the other, and a learner aims to select some number of whom. The authors propose a notion they term “Equalized Selection Rate (ESR)" which requires that the probability of an accepted qualified candidate is the same across groups in the distribution. The authors further suggest a post-processing approach for deriving a predictor that obeys the ESR constraint via solving a linear program (for fair selection using a binary classifier) or using a Bayesian optimization approach (for fair selection using qualification score). Additionally, the authors consider a case where for privacy concerns, the protected attribute is only accessible after noise is added (for predicted individuals), and show that perfect ESR is still attainable (when training the model on individuals for which the unnoisy protected attributes are included). The paper concludes with empirical evaluation of the proposed approaches.

**Limitations And Societal Impact:**

As stated in the main review, I would like to see a more elaborate discussion about the potential societal effects of the proposed ESR condition, and a discussion regarding taking a post-processing approach rather than an in-processing one.

**Main Review:**

The problem formulation studied in the paper is interesting. Rather than considering classification in its most general form, the authors focus on a sequential selection problem, inspired by various real-life scenarios. I appreciate this novel problem formulation and believe it is well-motivated and is worthy of further attention. The paper is well-written and is easy to follow and understand. In terms of novelty of the considered notion, to the best of my understanding, ESR is essentially equivalent to a condition regarding the true positive rates, weighted by the marginal group sizes for qualified individuals. The technical contributions are mainly through the suggested post-processing algorithms, which operate through a reduction to a linear programming problem or using a Bayesian optimization approach, and the extensions to the private case. This solution scheme seems elegant and easy to operationalize, however, as is the case with prior post-processing approaches (e.g. Hardt et al. 2016), it seems like the resulting accuracy-fairness trade-off could be sub-optimal.

In terms of technical contribution, there is no analysis or bounds regarding the accuracy loss that may result from taking the suggested post-processing approach for achieving the ESR notion, as opposed to an in-processing approach. I believe providing such bounds, even under additional assumptions, can significantly strengthen the results of paper. The authors do not have a discussion as for their choice to focus on a post-processing approach with a pre-trained classifier assumption rather than an in-processing approach. One could imagine, for example, attempting to cast the objective of accuracy under the ESR condition as a series of cost-sensitive optimization problems, as done in the reductions approach to fair learning (Agarwal et al. 2018) or one of the online variants.

The proposed ESR notion should be better motivated. For example, consider a case where the distribution consists of two groups, one with probability 0.9 and the other with probability 0.1. Assume further that the fraction of qualified individuals across both groups is 0.9. Under the ESR condition, it is necessarily the case where qualified candidates from the larger group are denied at a much higher rate. A discussion regarding the advantages and disadvantages of the proposed notion of fairness should appear in the paper.

Questions to authors:

1. Prior work has suggested a post-processing scheme for a different yet closely related fairness definition, equal opportunity, by solving a linear program (Hardt et al. 2016). Can you elaborate on the differences between the approach in the paper and previously suggested post-processing approaches, and especially Hardt et al. 2016, in terms of the difficulty of achieving EO vs. ESR and how does the different objective alter the required solution method?

2. In line 83, the paper states "Hardt et al. in [9] introduce a post-processing algorithm to find an optimal binary classifier satisfying equal opportunity". It is important to notice that the approach of Hardt et al. does not in general guarantee optimality of the equal opportunity predictor learned, but only in very specific cases (Woodworth et al. 2017). Optimality is rather with regards to the class of post-processed predictors, which can be far from general optimality under the equal opportunity condition. Can you discuss how taking a post-processing approach affects the resulting level of accuracy for the notion of ESR comparing to the optimal, not necessarily post-processed, solution over a given base class of predictors?

The submission appears to be technically sound, although I have only glanced at proofs in the appendix. The paper cites the relevant prior work.

**Time Spent Reviewing:**

8

---

> ### Author Response · Authors · 2021-08-10
> **Response to comments and addressing reviewer's concerns**
>
> $\textbf{Comment:}$ ESR is essentially equivalent to a condition regarding the true positive rates, weighted by the marginal group sizes for qualified individuals.
>
> $\textbf{Response:}$ Yes, you are right. this has been shown in Theorem 1.
>
> $\textbf{Comment:}$ Prior work has suggested a post-processing scheme for a different yet closely related fairness definition, equal opportunity, by solving a linear program (Hardt et al. 2016). Can you elaborate on the differences between the approach in the paper and previously suggested post-processing approaches, and especially Hardt et al. 2016, in terms of the difficulty of achieving EO vs. ESR and how does the different objectives alter the required solution method?
>
> $\textbf{Response:}$ Thank you for pointing this out. The post-processing method in Hardt et al. 2016 leads to a linear optimization problem. However, in selection problems, the post-processing scheme leads to a non-convex optimization problem. One of the contributions of this work is that we showed that under some $\textbf{mild assumptions}$, the non-convex problem can be reduced to a linear program.
>
> $\textbf{Comment:}$ The proposed ESR notion should be better motivated. For example, consider a case where the distribution consists of two groups, one with probability 0.9 and the other with probability 0.1. Assume further that the fraction of qualified individuals across both groups is 0.9. Under the ESR condition, it is necessarily the case where qualified candidates from the larger group are denied at a much higher rate. A discussion regarding the advantages and disadvantages of the proposed notion of fairness should appear in the paper.
>
> $\textbf{Response:}$ Thank you for the interesting example. Yes, it is true that ESR may lead to a higher rejection rate for the larger group. This is because the primary goal of ESR is to ensure the limited positions to be filled with a diverse group of people. Consider the same example where 100 applicants compete for 3 positions, among all applicants 90 are from group 0 while 10 are from group 1. Under equal rejection rate, it’s very likely that 3 positions are all filled with people in group 0 and zero from group 1, leading to a lack of diversity. As such, to improve the diversity when the number of selections is very limited (i.e., under ESR condition), people from different groups need to be rejected at different rates. Note that if we want to decrease the rejection rate for the larger group, we can use an approximate version of ESR.
>
> $\textbf{Comment:}$ In line 83, the paper states "Hardt et al. in [9] introduce a post-processing algorithm to find an optimal binary classifier satisfying equal opportunity". It is important to notice that the approach of Hardt et al. does not in general guarantee optimality of the equal opportunity predictor learned, but only in very specific cases (Woodworth et al. 2017). Optimality is rather with regards to the class of post-processed predictors, which can be far from general optimality under the equal opportunity condition. Can you discuss how taking a post-processing approach affects the resulting level of accuracy for the notion of ESR comparing to the optimal, not necessarily post-processed, solution over a given base class of predictors?
>
> $\textbf{Response:}$ Thank you for bringing this up. In our setting, the quality of the predictor $R$ determines the accuracy of the derived predictor $Z$. Consider the scenario in Section 3.1, assume $Z^* = r^*(X,A)$ is the binary predictor which obtains the general optimality under ESR notion (e.g., it can be achieved with in-processing methods). Further, assume $R$ is a given predictor and we want to find predictor $Z$ satisfying ESR using the post-processing method (optimization problem (4)). We have the following theorem:
>
>
>   $\textsf{Theorem:}$  If $|\Pr\\{Y=1|Z^*=1\\} -\Pr\\{Y=1|R=1\\} | <\epsilon$ and $\Pr\\{A=a|Y=1,R=1\\} = \Pr\\{A=a|R=1\\}, \forall a\in\\{0,1\\}$, then $|\Pr\\{Y=1|Z^*=1\\} -\Pr\\{Y=1|Z=1\\}  | <\epsilon$,
>
> where $\Pr\\{Y=1|Z^*=1\\}$ and $\Pr\\{Y=1|Z=1\\}$ are the accuracy of predictor $Z^*$ and $Z$, respectively. This theorem implies that under certain condition, if the accuracy of pre-trained model $R$ is sufficiently close to the accuracy of $Z^*$, then the accuracy of predictor $Z$ is also close to the accuracy of $Z^*$.
>
> We will add this theorem to the paper to demonstrate the impact of predictor $R$’s quality on the optimality of predictor $Z$.
>
>
> $\textsf{Proof sketch:}$  Without loss of generality, suppose $\Pr\\{R=1,Y=1,A=0\\} \leq \Pr\\{R=1,Y=1,A=1\\}$. Consider the following feasible point of optimization problem (4),
>
> $\hat{\alpha}_{00} =  0$
>
>  $\hat{\alpha}_{10} = 0$
>
>  $\hat{\alpha}_{01} = 1$
>
> $ \hat{\alpha}_{11} =\frac{\Pr\\{R=1,Y=1,A=0\\}}{\Pr\\{R=1,Y=1,A=1\\}} $
>
> Using these parameters, we can derive a new predictor $\hat{Z}$. Note that $\Pr\\{R=1|\hat{Z} = 1\\} = 1$ holds.  Since $\hat{Z}$ is a suboptimal solution to optimization problem (4) while $Z$ is the optimal solution to (4), We have
>
>  $$|\Pr\\{Y=1|Z^*=1\\} -\Pr\\{Y=1|Z=1\\}| \leq |\Pr\\{Y=1|Z^*=1\\} -\Pr\\{Y=1|\hat{Z}=1\\}|=$$ $$|\Pr\\{Y=1|Z^*=1\\} -\Pr\\{Y=1|R=1\\}| + |\Pr\\{Y=1|\hat{Z}=1\\} -\Pr\\{Y=1|R=1\\}| \leq$$ $$ \epsilon + |\Pr\\{Y=1|\hat{Z}=1\\} -\Pr\\{Y=1|R=1\\}| $$
>
>  Moreover, we have,
>  $$ \Pr\\{Y=1|\hat{Z}=1\\} = \Pr\\{Y=1|\hat{Z}=1,R=1\\} = \frac{\Pr\\{Y=1|R=1\\}}{\Pr\\{ \hat{Z}=1|R=1\\}} \Pr\\{\hat{Z}=1|Y=1,R=1\\}$$
>  $$\Pr\\{ \hat{Z}=1|R=1\\} = \Pr\\{A=0|R=1\\}+ \Pr\\{A=1|R=1\\} \hat{\alpha}_{11}$$
>
>  $$\Pr\\{\hat{Z}=1|Y=1,R=1\\} = \Pr\\{A=0|Y=1,R=1\\} + \Pr\\{A=1|Y=1,R=1\\}\hat{\alpha}_{11} $$
>
>   $$ \rightarrow |\Pr\\{Y=1|Z^*=1\\} -\Pr\\{Y=1|Z=1\\}| \leq \epsilon + \Pr\\{Y=1|R=1\\} (1-\frac{ \Phi_0 }{\Phi_1} )$$
>
> where $\Phi_0 :=\Pr\\{A=0|Y=1,R=1\\} + \Pr\\{A=1|Y=1,R=1\\}\hat{\alpha}_{11}$
>
> and $\Phi_1:=\Pr\\{A=0|R=1\\} + \Pr\\{A=1|R=1\\}  \hat{\alpha}_{11}$
>
>
> Since $\Pr\\{A=a|Y=1,R=1\\} = \Pr\\{A=a|R=1\\}, \forall a\in\\{0,1\\}$, we have $\Phi_0 =\Phi_1$. The theorem is proved.

---

> > ### Author Response · Authors · 2021-09-01
> > **Message to reviewer**
> >
> > Thanks again for your comments! We hope our response has addressed all your concerns. Since we are approaching the end of the discussion period, please let us know if you have any other questions, and we are happy to discuss more. If you’re satisfied with our response, we sincerely hope you could reconsider the rating.

---

> > > ### Comment · Reviewer_ffFj · 2021-09-02
> > > **Thank you for the response**
> > >
> > > Thank you for the response.
> > >
> > > One question for which I thought there was no clear answer in the rebuttal is regarding your choice of taking a post-processing approach, which could be sub-optimal, as opposed to an in-processing approach. The provided theorem shows optimality, but only under restrictive conditions. Could you elaborate further on the reasoning behind this choice of yours?

---

> > > > ### Author Response · Authors · 2021-09-02
> > > > **Response to the reviewer**
> > > >
> > > > Thank you for your comment.
> > > >
> > > > As we mentioned in the introduction, there are three main approaches to finding a fair ML model: pre-processing, in-processing, and post-processing. All of these approaches are valid solutions for bias mitigation in machine learning, and there are tons of studies on each of them. Each of these approaches has its own pros and cons.
> > > >
> > > > While our post-processing approach finds a near-optimal solution, it gives a simple and effective framework for fair selection. The method is computationally efficient and can use the output of a pre-trained machine learning model (not the feature vector $X$) to find a fair model. Rather than changing a possibly complex training pipeline as an in-processing approach does, the post-processing framework is easy to implement. This is why it has gained much attention.
> > > >
> > > > Also, we want to emphasize that the main message of our paper is not about the post-processing approach. Our main message is that the commonly used fairness notions such as equal opportunity and statistical parity cannot mitigate disparity in a selection problem where competition exists. In order to convey this message, we proposed a new fairness notion and adopted a post-processing framework to satisfy that. An in-processing approach for fair selection could also be an interesting problem and can be considered as potential future work.

---

> > > > > ### Comment · Reviewer_ffFj · 2021-09-02
> > > > > **Thank you for the response**
> > > > >
> > > > > Thank you for the response.

---

### Official Review · Reviewer_wyEe · 2021-07-19

**Rating:** 6
**Confidence:** 3

**Summary:**

This paper introduces a new fairness notion called Equalized Selection Rate (ESR) for sequential selection problems in which individuals are seen by the decision-maker sequentially, and there are a limited number of positive decisions available. For example, in a job acceptance setting, individuals are applicants submitting resumes sequentially, and the process ends when the decision-maker has accepted $m$ applicants. The authors show that a pre-trained model able to satisfy statistical parity or equal opportunity can be unfair in this sequential setting. A post-processing method is provided to circumvent this issue. Moreover, the authors show that when sensitive attributes are perturbed according to a local differential privacy notion, ESR fairness is still obtainable. Lastly, an empirical analysis is performed comparing equalized odds and ESR over two real data sets.

**Ethical Concerns:**

None.

**Limitations And Societal Impact:**

Yes.

**Main Review:**

originality -- I thank the authors for the extensive related work section---related work in classification fairness, selection, and differential privacy is discussed. Work related to sequential decision making is briefly mentioned (a single RL algorithm). To add, the bandit literature is another large area in sequential decision making with numerous works in fairness, e.g., Joseph et al. Fairness in Learning: Classic and Contextual Bandits, which focuses on individual fairness. More closely related to the classification setting (does not deal w/ exploration problem) might be batch bandit/RL algorithms that enforce fairness, e.g., Metevier et al. Bandits with High Prob. Guarantees. ESR is a fairness notion based on predicted outcome and not actual outcome (the selection made and not the "selection that should have been made"), and can be thought of as the action taken by the bandit. To summarize, along with reinforcement learning, I suggest mentioning bandit literature in the related work, as it is another important area in sequential decision making. Discussing the differences in the RL/bandit settings and the one being considered in this work can also strengthen the related work section.

clarity/quality -- The paper is readable, but there is some room for improvement. In line 183 "It shows that even with the seemingly fair..." this requires more explanation, e.g., an example or intuition. Line 147 offers good motivation for the need of a fairness definition for the sequential setting. Also, notation for probabilities seem to change in Theorem 2 (from brackets to parentheses). I did not view the supplementary material but looked at the work in the main text. Is there a reason the authors switch between parens and bracket notation for probability?

significance -- The authors assert in the introduction that they show that models enforcing statistical parity / equal opportunity may still be fair in the sequential setting they consider. I only see this in (1) after Corollary 1, line 183, "Note that he condition in Corollary 1 generally does not hold. It shows that...we may still be discriminatory,"  and (2) the comparison to equal opportunity in the experimental section. A comparison to statistical parity seems to be missing entirely in the empirical section of the main text, although it is listed as a contribution. Otherwise, the empirical analysis seems to support the claims made by the paper.

**Time Spent Reviewing:**

7

---

> ### Author Response · Authors · 2021-08-10
> **Response to comments and Addressing Reviewer's concerns**
>
> $\textbf{originality}$
>
> $\textbf{Regarding Multi-Armed Bandit (MAB) Problems:}$ We thank the reviewer for pointing this out. We will add the related MAB literature to the related work section.
>
> It is worth mentioning that sequential selection using a supervised learning problem is different from MAB problems. In this paper, our goal is to find a supervised learning model which is used sequentially. The decision-maker has access to the training data (or the distribution of the applicant’s data) and finds a supervised model before applicants arrive at the system. However, in MAB, there is no training data, and the learning process is through exploration and exploration algorithms: at each time slot, one (or more) applicant is selected, and the true qualification status (a.k.a reward in MAB problem) of the selected applicant is revealed immediately.  This may not be realistic in applications such as hiring, admission, etc., where it may take a long time for the decision-maker to observe the true labels of selected applicants (e.g., in college admission, it takes four years for an applicant to graduate.).
>
> We went through the suggested related literature (Joseph et al. Fairness in Learning: Classic and Contextual Bandits). This work considers the MAB problem with the $\textbf{individual}$ fairness notion, which is totally different from our work: we consider a supervised learning problem with a $\textbf{group}$ fairness notion.
>
> We also went through another work mentioned by the reviewer (Metevier et al. Bandits with High Prob. Guarantees). This work considers an offline bandit problem and does not model the competition among the applicants, i.e., no limit on the number of approvals.
>
> $\textbf{Clarity and Readability}$
>
> $\textbf{Regarding line 183:}$ Thank you for pointing this out. By seemingly fair, we meant the decisions that satisfy common fairness notions such as equal opportunity (EO) and statistical parity (see line 184). We have shown empirically that the equal opportunity fairness notion leads to an unfair outcome (Table 1 & 2). We also conducted another experiment to show that statistical parity also leads to an unfair outcome; the new numerical results are included below and will be added to the paper.
>
> To see why EO is not suitable for our problem, consider an example where 100 qualified applicants competing for 3 positions. Among them, 90 are from group 0 while 10 are from group 1. Under ESR, the probability that each position is filled with an applicant from group 0 is the same as that from group 1. Therefore, we expected that the selected applicants are diverse and coming from both groups. However, neither statistical parity nor equal opportunity cares about diversity, and they are likely to result in all the positions being filled by the majority groups. We will add this discussion to the paper.
>
>
> $\textbf{Is there a reason the authors switch between parens and bracket notation for probability?}$ Thank you for mentioning this. No, there is no reason. We will make them consistent and use the bracelet for probability.
>
> $\textbf{Significance}$
>
> $\textbf{More numerical results regarding the statistical fairness notion.}$ Thank you for pointing this out. We performed more numerical experiments about statistical parity using the FICO score dataset. We will add the following results to Table 1 ($\gamma$-SP implies that $|\Pr\\{Z=1|A=0\\} - \Pr\\{Z=1|A=1\\} |\leq \gamma$):
>
> Metric	$\hspace{0.5cm }$ $\tau_1$ $\hspace{0.5cm }$	$\tau_2$ $\hspace{0.2cm }$	 $\Pr\\{E_0, \\tilde{Y} = 1\\}$ $\hspace{0.2cm }$ $\Pr\\{E_1, \\tilde{Y} = 1\\}$$\hspace{0.2cm }$  Acc
>
> 0.001-SP	$\hspace{0.2cm }$ 99.5 $\hspace{0.2cm }$	99.5 $\hspace{0.5cm }$			0.9907		$\hspace{2cm }$		0	  $\hspace{1cm }$         0.9907
>
> 0.01-SP	$\hspace{0.4cm }$ 99.5  $\hspace{0.2cm }$	99.5 	$\hspace{0.5cm }$		0.9907	$\hspace{2cm }$			0	  $\hspace{1cm }$          0.9907
>
> Also, we will add the following rows to Table 2
>
> Metric	$\hspace{0.5cm }$	$\tau_1$ $\hspace{0.5cm }$	$\tau_2$	$\hspace{0.2cm }$ $\Pr\\{E_0, \\tilde{Y} = 1\\}$ $\hspace{0.2cm }$     $\Pr\\{E_1, \\tilde{Y} = 1\\}$ $\hspace{0.2cm }$   Acc
>
> 0.001-SP 	$\hspace{0.2cm }$	98.0 $\hspace{0.2cm }$94.0 	   $\hspace{0.5cm }$			0.8735	$\hspace{2cm }$		0.1146	 $\hspace{1cm }$	       0.9881
>
> 0.01-SP	 $\hspace{0.4cm }$     98.0 $\hspace{0.2cm }$	98.0 	$\hspace{0.5cm }$			0.9761		$\hspace{2cm }$	0.0138	 $\hspace{1cm }$	       0.9899
>
> These results show that statistical parity also leads to an unfair outcome in a sequential setting and the chance that an African-American applicant is selected is slim.

---

### Decision · Program_Chairs · 2021-09-27

**Decision:**

Accept (Poster)

**Comment:**

Reviewer all agreed that the paper formulates and partially addresses an interesting and practically relevant question: namely, how to define and guarantee an appropriate notion of fairness in sequential decision-making settings where a limited number of positions are available and diversity among the selected is an important consideration. However, among other suggestions for improvement, reviewers urge the authors to expand their discussion of the related work, the motivation behind proposed fairness notion, and the post-processing approach. Assuming that the authors will add the necessary discussions and revise the paper to reflect the reviewers’ suggestions, I recommend acceptance.